# Legal Origins and Corporate Social Responsibility

**Leonardo Becchetti** [1,*]**, Rocco Ciciretti** [1]  **and Pierluigi Conzo** [2,3]

1 Department of Economics and Finance, University of Rome Tor Vergata, 00135 Rome, Italy; rocco.ciciretti@uniroma2.it
2 Department of Economics and Statistics "Cognetti de Martiis", University of Turin, 10155 Turin, Italy; pierluigi.conzo@unito.it
3 Collegio Carlo Alberto, 10122 Turin, Italy
* Correspondence: becchetti@economia.uniroma2.it

**Abstract:** The legal origin literature documents that civil and common law traditions have different impacts on economic outcomes. We contribute to this literature by formulating and testing hypotheses on the effect of legal origins on corporate social responsibility, overall and in different specific dimensions. We find that, net of industry-specific effects, companies in common law countries score higher in corporate governance and community involvement, while those in countries belonging to the French legal tradition of civil law do better in human resources. We also observe no significant differences in terms of environmental protection among companies in civil and common law countries, which we attribute to a progressive convergence towards common industry sustainability standards.

**Keywords:** legal origins; corporate social responsibility; common law; environmental standards; corporate governance; civil law

## 1. Introduction

Advertising social and environmentally friendly behavior, issuing sustainability reports, and hiring Corporate Social Responsibility (CSR, hereon) experts has increasingly become corporate practice in the most recent years [1]. The growing relevance of CSR is leading academicians to reflect on whether the latter represents a major change in the economic paradigm with respect to the standard approach. In this approach, forces of market competition transform individual and corporate self-interested behavior into an efficient and socially optimal outcome, while the state intervenes with taxes and regulation to address the problem of externalities and public goods redistributing income and wealth according to the dominating social standards [2]. In this framework, the invisible hand of the market and the "visible" hand of institutions are sufficient to ensure socially optimal outcomes without the need for an explicit voluntary corporate effort toward social responsibility.

The demand for CSR has emerged mainly in recent decades. CSR was an almost irrelevant issue in pre-globalization high-income economies, where domestically producing firms already strived in high-income countries to comply with demanding domestic social and environmental rules and did not have much room for additional voluntary compliance to above the law standards. Quite to the contrary, in globally integrated economies in which production is delocalized in countries where legal standards are low, the role of CSR is becoming progressively more important in the eyes of consumers, domestic institutions, and investors [3]. We implicitly refer here to the EU Commission [4] definition of CSR as "a concept whereby companies integrate social and environmental concerns in their business operations and in their interaction with their stakeholders on a voluntary basis". From a different point of view, based on standard CSR measures (see also Appendix B), we may consider CSR as a move from the goal of maximizing shareholders' wealth to the more complex goal of satisfying the wellbeing of a broader range of stakeholders.

According to the Global Sustainable Investment Alliance [5], socially responsible investments totaled $30.7 trillion, an increase of 34% from 2016 and account for 18% of total assets under management in Japan, 63% in Australia and New Zealand, and 49% in Europe Exclusionary screens remains the dominant strategy with €9.5 Trillion. A global survey on consumers documents that the share of respondents saying that it is extremely important for companies to implement programs to improve the environment is 72% among baby boomers and 85% among millennials, with a geographical distribution showing higher sensitivity in Asia, Africa, and Latin America with respect to Europe and the United States [6]. Even though the willingness to pay for CSR tends to be upward biased, as documented by the contingent evaluation literature [7], these figures reveal that the phenomenon is quantitatively relevant.

As is well known from a theoretical point of view, adoption of CSR entails extra costs to satisfy the needs of stakeholders, which can be compensated by several potential benefits related to productivity of intrinsically motivated workers, reduced turnover, development of technological leadership in environmental innovation, lower transaction costs with stakeholders and higher demand from concerned consumers. This is why the empirical literature finds mixed results on the nexus between CSR and corporate performance [8]. However, while most of the literature has focused so far on the nexus between CSR and corporate performance, few empirical contributions analyze nowadays how different legal cultures affect CSR choices around the world (see [9] and [10], among others). This is the goal of our paper.

If markets are "embedded" in human societies and shaped by their socio-political contextual features [11], a relevant benchmark reference to start our investigation on the nexus between CSR and different country cultures is the legal origin literature. In their survey of this literature, La Porta et al. [12] argued that historical origins of domestic legal systems deeply affect legal rules, regulatory practices, and economic outcomes. As is well known, they identified two main roots, namely civil and common law, giving birth to four legal families, i.e., the Anglo-Saxon for the common law and the French, the German, and the Scandinavian for the civil law.

From an historical point of view, common law is generally considered as taking origin from the desire of land aristocrats and merchants to limit the power of the crown, while the French version of civil law from the Napoleon desire to "use state power to alter property rights" and in an attempt "to insure that judges did not interfere" [12]. Due to these heterogeneous historical roots, two markedly different cultures originated from civil and common law, with state control prevailing in the first, while support to private outcomes in the second [13]. According to Hayek [14], the two cultures imply two different conceptions of freedom: a freedom "from" and "of" for the common law, against a freedom "for" in the civil law, where social goals inspire the system of law and regulation. Using Djankov et al.'s [15] expression, in the dilemma between addressing market failure with regulation and avoiding state abuse, civil law is more oriented toward the former and common law toward the latter. This explains why civil law is "policy implementing", while common law is "dispute resolving" [16].

La Porta et al. [12,17] also showed that the two different cultures produce significant disparities in terms of rules and economic outcomes. Common law countries generally have higher shareholder and creditor protection and more capitalized stock exchanges than civil law countries. The latter are also shown to have higher government ownership and regulation than the former, which are characterized in turn by greater independence of the judicial power with better contract enforcement as well as security of property rights.

However, the study by La Porta et al. [12] has been criticized for lacking a proper theoretical channel through which the common law tradition provides more protection to non-controlling shareholders against insiders [18]. Moreover, Rajan and Zingales [19] and Lamoreux et al. [20] provided historical evidence on the relative development of capital markets in civil/common law countries with specific reference to the France-UK comparison. As a partial answer to the above critiques, Beck et al. [21] identified two patterns which *de facto* explain the observed correlation between common law and shareholder protection. The first is the priority given to the right of the individual *vis-à-vis* the state. The second is the higher flexibility with respect to more rigid financial traditions in reducing the gap

between needs of the economy and legal system capability to foster financial development. These two patterns may actually reconcile the perspective of La Porta et al. [12] and the empirical evidence accounting for both the higher correlation between common law and shareholder protection today and historical phases in which some civil law countries such as France may have performed relatively better in terms of financial market development.

Also the international management literature has investigated the relationships between institutional features and CSR. Matten and Moon [22] and Aguilera et al. [23] pioneered theoretical frameworks to explain differences in CSR across countries. In the same spirit of La Porta et al. [12], the former argued that these differences can be due to historically different institutional frameworks that shaped "national business systems" [24]—common law countries tend to favor "explicit" CSR policies, while civil law countries would foster "implicit" CSR ones. The first case refers to companies that explicitly and voluntarily implement CSR policies, whereas in the second case CSR is embedded in the national formal and informal institutions under the form of "[ . . . ] values, norms, and rules that result in (mandatory and customary) requirements for corporations to address stakeholder issues and that define proper obligations of corporate actors in collective rather than individual terms [ . . . ]" (Matten and Moon [22] p. 409).

According to Aguilera et al. [23], since corporations are embedded in different national business systems, they face different internal and external pressures to implement CSR policies. In particular, they argued that shareholders in the Anglo-American model would support CSR activities, leading to short-term benefits, whereas shareholders in Continental Europe would focus on long-term benefits for a larger set of stakeholders. Institutional features specific to the two national systems would explain the differences in CSR policies between the two models—e.g., dispersed ownership focusing on short-term returns within the Anglo-American model versus long-term debt finance and ownership of large shareholders within the European model.

The aim of our paper is to give a contribution to this literature by investigating the nexus between legal origins and CSR [13] in order to understand: i) why some companies engage more in CSR and why others do not; ii) whether this is due to cultural factors and, more specifically, legal origins; iii) what consequences the different emphasis on specific CSR domains have on the corporate strategy and wellbeing of different stakeholders, i.e., workers, local communities (through stronger emphasis on environmental sustainability), and shareholders (with stronger emphasis on corporate governance). In terms of policy implications, our research offers important insights on the extent to which the CSR stance affects the way societies deal with externalities and/or provide public goods, and on whether the latter are addressed/provided more by regulation or governmental action as opposed to voluntary corporate action.

Consistent with the aforementioned theoretical framework, we formulate hypotheses on how different institutional features rooted in different legal origins translate into different CSR domains, and test them econometrically. More specifically, since common law countries tend to have higher corporate discretion and disclosure of information, more efficient financial markets and higher transparency and accountability than civil law ones, they are traditionally more oriented at protecting shareholders' interests and therefore might score higher in the corporate governance domain. Moreover, we also hypothesize that common law countries have higher CSR scores in the community involvement domain, because of the "two-step" Anglo-Saxon culture—whereby charitable contributions are accepted after satisfying profit maximization—and the higher philanthropy capacity. Finally, we expect higher CSR scores of civil law countries in the labor domain (Human Resources), as those countries enjoy a more generous legislation in favor of workers and less shareholder (more stakeholder) protection than common law ones. We also document the unexpected finding of a lack of observable differences in the environmental domain, which we attribute to a progressive convergence from different legal systems to industry-sustainability standards, which are increasingly being adopted worldwide.

The paper is divided into six sections. In the second and third we formulate our theoretical hypotheses by analyzing how and whether a given legal tradition may be expected to be more favorable

to compliance toward a given CSR domain. In the fourth we present our data. In the fifth we illustrate our descriptive and econometric findings. The sixth section concludes.

## 2. Legal Origin Culture and Stakeholder Rights

The classic view of the legal purpose of the corporation originated in common law countries—*shareholder primacy*—seems to leave almost no room for that part of corporate social responsibility intended as a departure from profit maximization toward the satisfaction of a broader range of stakeholders [25–28]. According to this view, the manager receives a mandate from their employees (the shareholders) to maximize the profits of the company in the respect of the law. In this perspective, CSR entails the risk that the manager abuses of their own power to perform actions that waste corporate cash flow and are directed to increase their prestige beyond the screen of promoting the wellbeing of the other corporate stakeholders.

A view that is quite similar in its consequences to that of shareholder primacy defines the company as a *nexus of contracts* between suppliers and various production factors [29,30]. These contracts ensure that factors of production receive a fixed payment in exchange for their services, while shareholders are residual claimants of all the remaining cash flow. Similar to the shareholder primacy view, the *nexus of contracts* view regards any reduction of the shareholder residual as something that is unfairly subtracted from their pockets. The legitimacy of the shareholder claim on the residual cash flow is generally based on the idea that shareholders are those who bear most of the risk in the corporate venture, since their remuneration is more volatile than the fixed payment due to workers.

A third novel view of the legal purpose of the corporation was developed by Blair and Stout [31]. By criticizing this last point, the authors placed emphasis on the fact that resources invested by shareholders (money) are much more diversifiable than those invested by suppliers and workers (their skills and human capital) in the venture. As a matter of fact, in case of corporate failure, a shareholder with a well-diversified portfolio of financial assets may suffer fewer negative consequences than middle-aged low-skilled workers who invested all in job skills which may have become obsolete after corporate failure. This is one of the reasons why Blair and Stout [31] defined the company as a team, finding it therefore reasonable that the company uses the value added it produces to remunerate stakeholders in proportion to their merit and contribution. This third view is obviously much more favorable to non-shareholder-oriented CSR domains than the previous two.

Reinhardt's [32] conclusion on the US view of the legal purpose of the company is that the first two approaches (shareholder primacy and firm as a nexus of contracts) remain prevalent. What is, however, noted is that a "two-step" approach to CSR, where many states recognize the right of businesses to make charitable contributions after satisfying profit maximization, is quite popular in the US and, more in general, in Anglo-Saxon countries. This tradition of corporate philanthropy traces back to the well-known examples of Andrew Carnegie, John D. Rockefeller, and Henry Ford, among others. Nevertheless, a weakness of the shareholder primacy tenet in the US is that courts are generally quite indulgent toward managerial behavior. This is because they admit that it is difficult to bridge the informational asymmetry toward managers to establish "second guesses" beyond their actions that were not directed to the benefit of corporate profits. Last, but not least, many US jurisdictions have adopted "non-shareholder constituency statutes", which mitigate the shareholder primacy principle [33].

In conclusion, in spite of the prevalence of the first two views, which are quite hostile to non-shareholder-oriented CSR, the "two-step" tradition of corporate philanthropy and the indulgence of tribunals lead us to expect a development of CSR in the US (and more in general in Anglo-Saxon countries) also beyond traditional corporate governance rules protecting shareholders, and especially in the direction of monetary donations to local communities.

On the opposite side, it is reasonable to expect that attitudes toward CSR in civil law countries reflect characteristics described in the legal origin theory—generally lower shareholder protection and a cultural *milieu* in which economic activity must be oriented toward social goals (which mitigates

shareholders' interest and aim to increase the wellbeing of other stakeholders). In this sense, civil law countries have developed a tradition that is much closer to Blair and Stout's [31] idea of the corporation as a team, whereby the added value generated by the creativity of corporate activities must be redistributed across different shareholders, with the board of directors balancing the competing demands of team members (stakeholders). This view is supported by Roe [34], who argued that in countries such as Germany and France stakeholders (and in particular, employees) have much stronger legal power than in the United States. This different attitude may be fostered also by differences in shareholders' ownership, whereby a few large shareholders may be more likely (and have more power) to commit socially than the dispersed shareholders of US companies.

## 3. Our Research Hypotheses

The main question we aim to answer in this paper is which CSR domains are correlated with civil and which with common law, or, more broadly, with the four families of legal origins (Anglo-Saxon, French, Scandinavian, and German). Based on the literature surveyed above we expect the following:

i)   Common law countries have higher CSR scores in the corporate governance domain (which is traditionally oriented to promote shareholders' wellbeing);
ii)  Common law countries have higher CSR scores in the community involvement domain;
iii) Civil law countries (and, more specifically, the French tradition) have higher scores in the CSR labor domain (human resources).

As should be clear from what was discussed in the previous sections, these three hypotheses stem from the characteristics outlined by the legal origin literature when applied to a relatively novel investigation field such as that of CSR.

In particular, hypothesis i) is supported by the idea that common law countries are characterized by higher corporate discretion and disclosure of information [12,17,35], as well as an efficient financial source based on a central stock market [22], with low shareholder dispersion leading to higher transparency and accountability [36–38]. Conversely, in civil law countries—characterized by concentrated ownership, underdeveloped financial markets, and low transparency and accountability [22]—stakeholders other than shareholders also play a role that sometimes is even more important than that of shareholders [39].

Hypothesis ii) is derived from the aforementioned two-step culture of profit maximization, followed by philanthropic donations typical of the Anglo-Saxon culture, as well as from the "explicit" CSR type of common-law countries, which enjoy a relative capacity for philanthropy [40], as theorized by Matten and Moon [22].

The third hypothesis is based on the cultural traditions of civil law countries, where laws in favor of workers are higher and shareholder protection lower. Coordinated market economies stimulate employee cooperation and the development of collective agreements in wage determination [41] and [24]. Moreover, in countries with a large role for government in the economy, distributional outcomes play an important role and tend to favor employees over capital-owners in case of conflict between the two [34]. CSR activities concerning human resources in civil law countries are more likely to be "implicit", since employee representation and participation are protected by an intensive employment regulation [22].

An important issue related to our hypotheses arises when we consider one of the most common CSR definitions, which relies on the integration of social and environmental concerns on a *voluntary* basis. We clarify here that what we measure with standard CSR rating agencies data (i.e., those by VIGEO) is a sort of gross CSR—that is, the total corporate engagement in a given domain, which may include legal standards required by a given country plus the net CSR component generated by corporate voluntary conduct. In this sense legal origins are expected to influence both components, since they affect not only legal rules but also culture, which may in turn impact corporate voluntary choices.

## 4. Data

Our sample period was from 2003 to 2013 and included 1834 unique companies. The dataset was created by merging five different sources: (i) data on CSR scores at company level from the VIGEO world dataset; (ii) stock prices from DATASTREAM; (iii) information on legal origins taken from La Porta et al. (2008); (iv) industry specification from Standard Industry Classification Code (SIC); and (v) controls at firm level from COMPUSTAT Global.

The firms included in the VIGEO world dataset were chosen amongst the components of the STOXX Global 1800 Index, with a preference for companies that were in the MSCI World as well. The VIGEO dataset covers 95.3% (65%) of the STOXX Global 1800 Index in terms of market capitalization (number of companies). Other companies were included if they were components of specific national indices (i.e., S&P/MIB for Italy). The STOXX Global 1800 Index provides a broad investable representation of the world's developed markets of Europe, North America, and Asia/Pacific. This Index contains 600 European, 600 American, and 600 Asia/Pacific region stocks represented by the STOXX Europe 600 Index, the STOXX North America 600 Index, and the STOXX Asia/Pacific 600 Index. Stocks were selected based on their rankings by free float market capitalization. Instead of using all the outstanding shares as in the full-market capitalization method, the free-float method excluded locked-in shares such as those held by promoters and governments. The number of companies in our dataset was higher than 1800 due to new entries in the STOXX global index during our sampling period.

VIGEO assesses CSR performance on six domains: human resources, environment, business behavior, corporate governance, community involvement, and human rights. Each domain is divided into a different number of drivers $k$, for a total of 38 drivers over six domains. According to industry $j$ in which firm $i$ belongs, drivers can be activated or not. Details on each domain and sustainability driver are provided in Appendix B.

The score for each domain $d$—$FS_d(ij)$ for firm $i$ in industry $j$ is computed as follows:

$$FS_d(ij) = \frac{\sum_{k=1}^{K} n_{ijdk} w_{ijdk}}{W_{jd}}, \tag{1}$$

where $n_{ijdk}$ is the score assigned in the driver $k$ to the firm $i$ belonging to the industry $j$, which goes from 0 to 100; $w_{ijdk}$ is the weight (going from 1 to 3) assigned in the driver $k$ to the firm $i$ belonging to the industry $j$; $W_{jd}$ is the sum of all the categories' weights activated in the domain ($W_{jd} = \sum_{k=1}^{K} w_{ijdk}$).

The value assigned depends from the difficulty in implementing CSR standards in each specific category than it is industry specific. If $w_{ijdk} = 1$, for the industry $j$ it's relatively easy to implement CSR standards in the category under analysis; vice versa for the case $w_{ijdk} = 3$. For example, the values taken by $w_{ijdk}$ tend to be low for a bank in the environmental domains, since respect of environmental standards is relatively easier for this industry. Note that $n_{ijdk} = 0$ indicates the lack of transparency of the firm in the category under consideration or the simultaneous presence of litigations; the opposite occurs in the case of $n_{ijdk} = 100$.

The weighted sum across all the domains is defined as the overall score for firm $i$ in industry $j$ —$OA_{ij}$—as follows:

$$OA(ij) = \frac{\sum_{d=1}^{D} FS_d(ij) \, W_{jd}}{W_j}, \tag{2}$$

where $W_j$ is the sum of all the categories' weights activated in all the six domains ($W_j = \sum_{d=1}^{D} \sum_{k=1}^{K} w_{ijdk}$).

Equation (1) measures the scores across $k$ drivers in a specific domain $d$, while Equation (2) measures across $k$ categories over all the six domains considered.

In order to control for the robustness of our results to the VIGEO weighting approach, we introduced fixed industry effects as regressors in our econometric estimates and, alternatively, we calculated scores as deviations from industry averages in the empirical analysis presented in the next sections.

## 5. Descriptive Statistics

Strictly following La Porta et al.'s [22] classification, we divided the firms in our sample according to the legal origins of their country; more specifically, we classify them in four categories, namely French, German, Scandinavian, or English legal origin. Firms in countries belonging to the first three categories were also grouped into the broader civil law category, while those belonging the fourth were grouped into the common law category (see Table 1 for country allocation to the different families).

**Table 1.** Classification of countries by legal origins.

| COMMON LAW | CIVIL LAW | | |
| English | French | Scandinavian | German |
|---|---|---|---|
| Australia, Canada, Hong-Kong, Ireland, New Zealand | Belgium, France, Portugal | Denmark, Finland, | Austria, Bermuda, China, Luxembourg |
| Singapore, United Kingdom, United States | Greece, Italy, Spain, Netherlands, Russia | Sweden, Norway | Germany, Iceland, Japan, Switzerland |

In Table 2 we report summary statistics for all the variables used in descriptive and econometric analysis. Civil and common law groups were almost balanced (52 against 48 percent), while the French and the German families were much larger within the civil law origin (each of them accounting for 23 percent of the overall sample against 7 percent of the Scandinavian family).

**Table 2.** Descriptive statistics.

| Variable | Obs. | Mean | Std. Dev. | (95% Conf. Interval) | | Min | Max |
|---|---|---|---|---|---|---|---|
| CSR score | | | | | | | |
| Overall score | 7000 | 36.064 | 12.301 | 35.378 | 36.751 | 4 | 77 |
| Human resources | 8137 | 28.990 | 17.646 | 28.029 | 29.951 | 0 | 84 |
| Environment | 8137 | 31.352 | 18.381 | 30.357 | 32.348 | 0 | 87 |
| Business behaviour | 8137 | 38.892 | 13.275 | 38.281 | 39.504 | 4 | 82 |
| Corporate governance | 8137 | 46.239 | 17.078 | 45.392 | 47.087 | 1 | 94 |
| Community involvement | 8137 | 36.064 | 18.547 | 35.086 | 37.041 | 0 | 96 |
| Human rights | 7000 | 39.391 | 14.422 | 38.619 | 40.163 | 3 | 91 |
| Legal origin | | | | | | | |
| Civil law | 8135 | 0.520 | 0.495 | 0.491 | 0.548 | 0 | 1 |
| English | 8137 | 0.480 | 0.495 | 0.452 | 0.509 | 0 | 1 |
| French | 8137 | 0.229 | 0.453 | 0.203 | 0.255 | 0 | 1 |
| German | 8137 | 0.225 | 0.411 | 0.201 | 0.248 | 0 | 1 |
| Scandinavian | 8137 | 0.066 | 0.247 | 0.053 | 0.080 | 0 | 1 |
| Other variables | | | | | | | |
| Total Assets (ln) | 7749 | 0.076 | 0.632 | 0.044 | 0.108 | 0 | 21.834 |
| GDP per capita PPP (Purchasing Power Parity)/1000 | 8135 | 38.857 | 8.095 | 38.451 | 39.263 | 6.781 | 81.104 |
| G/GDP | 8135 | 43.782 | 7.048 | 43.391 | 44.173 | 14.432 | 64.902 |

In order to have descriptive evidence to test our hypotheses, we provide in Table 3 a breakdown of CSR criteria by legal origins. More specifically, when we considered simple CSR means, firms in civil law countries displayed higher CSR scores than those in common law countries under all criteria but corporate governance (all the differences were significant under parametric tests). Non parametric tests provide results which are not qualitatively different (available upon request).

**Table 3.** Corporate Social Responsibility (CSR) by legal origins.

| A) Mean CSR score by legal origin | | | | | | | |
|---|---|---|---|---|---|---|---|
| Legal Origin | Overall score | Human resources | Environment | Business behavior | Corporate governance | Community involvement | Human rights |
| ENGLISH | 35.794 | 24.554 | 29.842 | 39.150 | 55.757 | 36.996 | 37.298 |
| | (10.989) | (14.282) | (17.874) | (12.663) | (13.002) | (18.398) | (12.625) |
| FRENCH | 40.097 | 43.340 | 39.334 | 44.560 | 43.557 | 44.566 | 44.625 |
| | (12.856) | (16.052) | (17.369) | (13.396) | (13.819) | (17.556) | (15.326) |
| SCANDINAVIAN | 36.845 | 35.269 | 34.771 | 40.741 | 44.399 | 32.718 | 42.855 |
| | (11.229) | (15.959) | (18.285) | (13.167) | (12.381) | (17.689) | (14.847) |
| GERMAN | 32.120 | 30.363 | 34.356 | 38.013 | 32.581 | 33.266 | 37.501 |
| | (13.320) | (18.088) | (18.757) | (13.156) | (18.324) | (17.953) | (15.362) |
| COMMON LAW | 35.800 | 24.555 | 29.849 | 39.162 | 55.763 | 36.998 | 37.303 |
| | (10.989) | (14.284) | (17.877) | (12.656) | (13.002) | (18.395) | (12.624) |
| CIVIL LAW | 36.239 | 37.494 | 36.923 | 41.639 | 39.495 | 38.922 | 41.327 |
| | (13.387) | (17.908) | (18.173) | (13.625) | (16.451) | (18.618) | (15.644) |
| B) Test of mean CSR score by legal origin (*t*-statistics) | | | | | | | |
| Common versus Civil Law | −3.9544 *** | −15.7723 *** | −6.9578 *** | −5.7715 *** | 17.3054 *** | −2.1211 *** | −8.9743 *** |
| English versus French | −4.9787 *** | −17.3151 *** | −7.2738 *** | −6.8656 *** | 15.2730 *** | −6.2778 *** | −8.3713 *** |
| English versus German | −1.3065 * | −9.7134 *** | −5.4665 *** | −2.664 0*** | 15.4367 *** | 0.8974 | −6.0205 *** |
| English versus Scandinavian | −2.6224 *** | −10.1105 *** | −3.4363 *** | −3.5419 *** | 9.7071 *** | 3.2803 *** | −7.7834 *** |
| French versus German | 3.2572 *** | 5.5955 *** | 0.6951 | 3.7528 *** | 3.0236 *** | 6.9431 *** | 1.5557 * |
| French versus Scandinavian | 1.6488 ** | 3.9014 *** | 2.4252 *** | 2.0189 *** | −2.8465 *** | 8.7892 *** | −0.8435 |
| German versus Scandinavian | 1.3130 * | 1.0514 | −1.5323 * | 1.1957 | 4.9836 *** | −2.4288 *** | 2.0718 ** |

Std. dev. are reported in parentheses; * Significant at the 1% level; ** Significant at the 5% level; *** Significant at the 10% level.

The higher performance of civil law countries seems to be driven by countries with French legal origins, which enjoy higher CSR scores in all criteria relative to those with English legal origins, with the only exception being the corporate governance criterion, under which the latter performed better than the former.

This preliminary statistical testing was oriented toward the non-rejection of our hypotheses i) and iii), since common law countries exhibited better CSR performance in the corporate governance domain, whereas civil law countries in the human resources and human rights domains. Evidence on hypothesis ii) was instead contrary to what we expected.

However, the comparison of CSR means can be misleading if we do not consider the potential source of heterogeneity coming from industry-specific characteristics (which the VIGEO's weighting approach described above tries to address). To control for this, we calculated in each period and domain deviations of firm CSR scores from the average of the industry it belongs to (i.e., $FS_d(ij) - \sum_j FS_d(ij)$). The effect of industry-specific characteristics was further controlled for in econometric estimates, where dummies for industry were included among regressors. The descriptive analysis of industry deviation scores differs from the previously described patterns (Table 4). In particular, for the overall CSR score, civil law countries were, on average, below the mean industry overall CSR score.

This finding might appear in principle inconsistent with the high overall CSR-performance of civil law countries reported in Table 3. The combination of the two results suggests that civil law countries have a higher number of firms in CSR-friendly sectors than common law countries. The difference between the two results, instead, confirms the importance of industry characteristics, which we consider in the econometric estimates below.

However, and consistently with the previous results, civil law countries performed better (i.e., were above the industry average) under the human rights and human resources criteria, whereas the common law ones were in general above the industry average in all the other domains (most differences were significant in parametric tests).

This last piece of evidence gives additional support to our hypotheses i) and iii) and leads us also to not have any more mixed findings on hypothesis ii), since common law countries exhibited higher CSR performance also in the community involvement domain after netting out industry-specific characteristics (i.e., are above the industry CSR average in that domain).

**Table 4.** Deviations from industry average CSR by legal origins.

| Legal Origin | Overall score | Human Re sources | Environment | Business Be havior | Corporate Governance | Community In volvement | Human Rights |
|---|---|---|---|---|---|---|---|
| **A) Mean deviations from industry average CSR by legal origin** | | | | | | | |
| ENGLISH | 1.095 | −0.895 | 0.459 | 0.283 | 5.869 | 1.221 | −0.225 |
|  | (8.317) | (9.756) | (13.344) | (9.811) | (12.026) | (13.776) | (9.685) |
| FRENCH | −0.007 | 2.056 | −0.622 | 0.294 | −3.879 | 0.692 | 0.986 |
|  | (10.152) | (12.673) | (13.076) | (10.207) | (12.303) | (14.332) | (12.523) |
| SCANDINAVIAN | −2.215 | −1.206 | −1.316 | −0.808 | −4.033 | −5.945 | 0.156 |
|  | (9.766) | (12.728) | (14.113) | (11.138) | (11.360) | (15.152) | (12.741) |
| GERMAN | −1.650 | −0.587 | 0.311 | −0.714 | −5.351 | −1.566 | −0.575 |
|  | (9.206) | (11.625) | (13.744) | (9.547) | (11.572) | (12.520) | (11.230) |
| COMMON LAW | 1.100 | −0.894 | 0.468 | 0.295 | 5.863 | 1.224 | −0.220 |
|  | (8.316) | (9.755) | (13.342) | (9.801) | (12.027) | (13.777) | (9.684) |
| CIVIL LAW | −1.002 | 0.679 | −0.348 | −0.215 | −4.454 | −0.927 | 0.206 |
|  | (9.744) | (12.371) | (13.463) | (10.087) | (11.943) | (13.930) | (12.032) |
| **B) Test of mean deviations from industry average CSR by legal origin (*t*-statistics)** | | | | | | | |
| Common versus Civil Law | 3.9577 *** | −4.2117 *** | −0.4823 | −0.2473 | 20.2065 *** | 3.4607 *** | −1.9829 ** |
| English versus French | 2.5585 *** | −4.9788 *** | −0.2827 | −0.8787 | 18.0384 *** | 0.4510 | −1.7040 ** |
| English versus German | 4.0470 *** | −2.2191 ** | −1.0706 | 0.3612 | 15.8392 *** | 3.8860 *** | −0.9515 |
| English versus Scandinavian | 3.4012 *** | −2.4585 *** | −0.2366 | −0.3054 | 11.9200 *** | 6.6926 *** | −2.7290 *** |
| French versus German | 1.4541 * | 2.5351 *** | −0.8375 | 1.1751 | −0.7165 | 3.2695 *** | 0.7045 |
| French versus Scandinavian | 1.1658 | 1.5031 * | −0.0149 | 0.3781 | −2.7185 *** | 6.3368 *** | −1.1678 |
| German versus Scandinavian | 0.1050 | 0.6110 | −0.6567 | 0.5695 | 2.1098 * | −3.5324 *** | 1.7726 ** |

Std. dev. are reported in parentheses; * Significant at the 1 % level; ** Significant at the 5 % level; *** Significant at the 10 % level.

Last but not least, in order to analyze how corporate social responsibility has changed across the years, we display in Figure 1 (panels a–g) the time dynamics of average CSR scores. These figures reveal marked convergence pattern across different legal origin areas in the overall CSR score and particularly in the environment domain, where the two groups converged to a mean sample value (Figure 1, panel c). This descriptive evidence induced us to test econometrically the convergence hypothesis in the next sections.

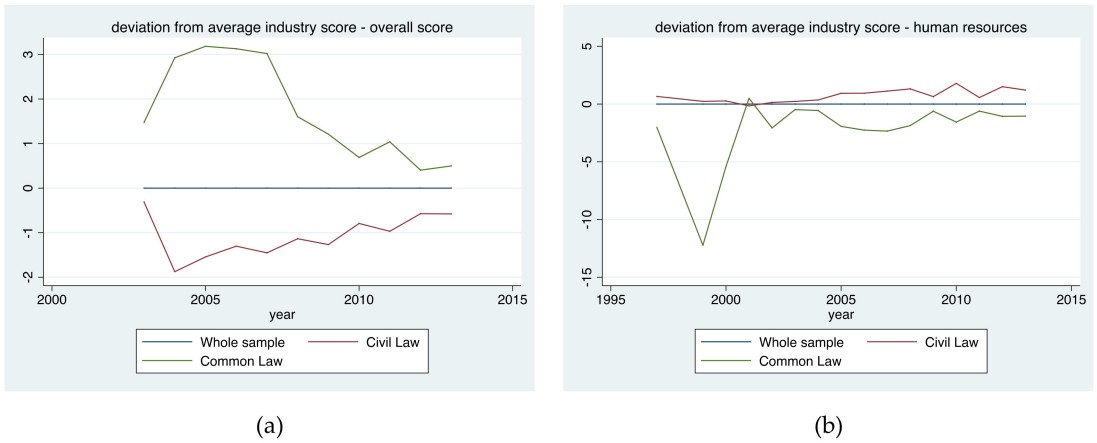

(a)                                                  (b)

**Figure 1.** *Cont.*

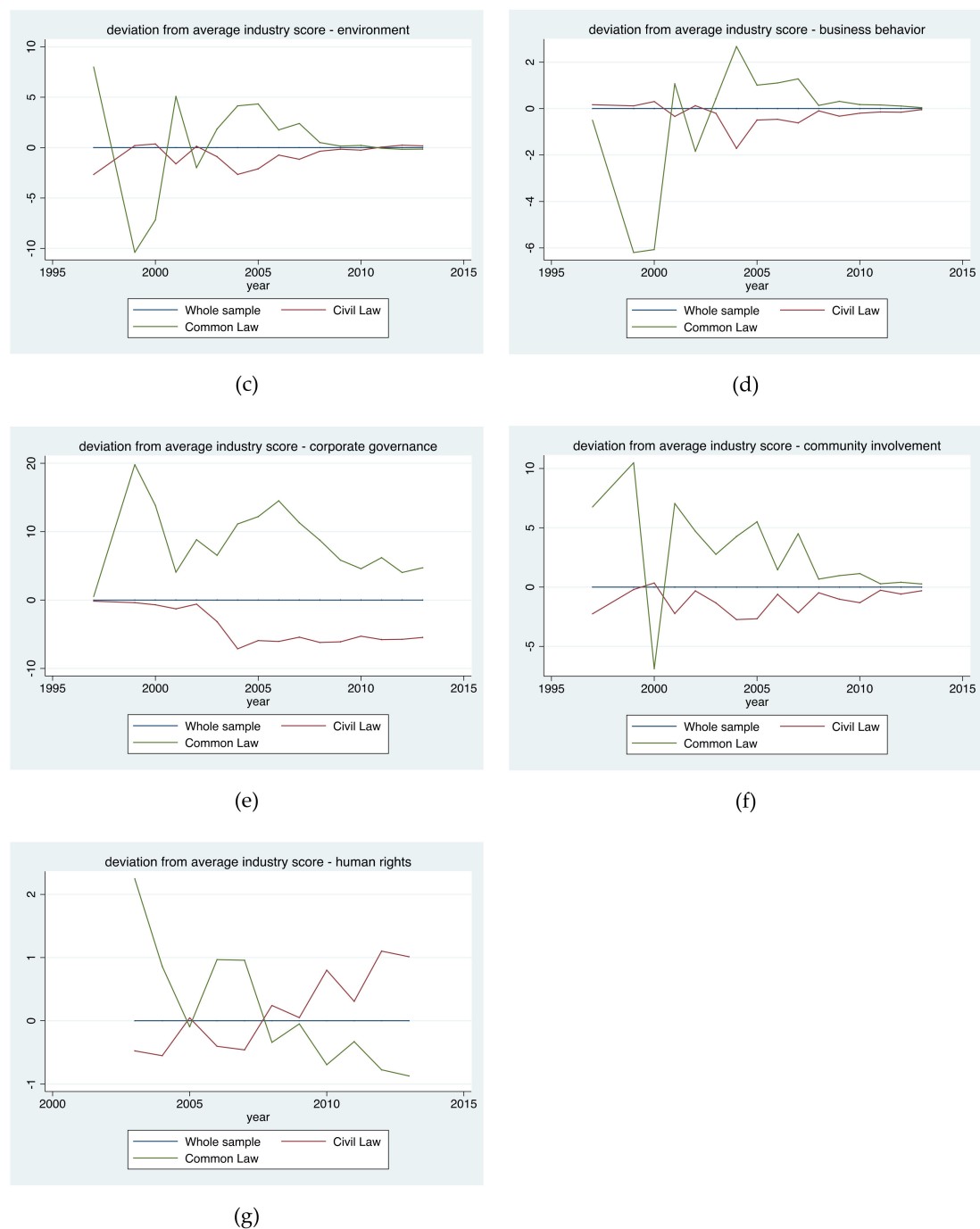

**Figure 1.** Deviations from industry average CSR over time.

## 5.1. Econometric Model and Results

The descriptive statistics and parametric tests presented so far did not take into account the panel structure of our dataset and did not allow us to isolate the impact of legal origin from other time, industry, and country specific characteristics that are also expected to influence CSR scores. Moreover, even though we have no reason to doubt that the VIGEO's weighting system reflects an expert intervention on a scoring process, which is in any case subjective and arbitrary also before the weighting intervention, we were interested in checking whether our main results were robust to a weakening of such weighting effect. For these reasons, we ran an econometric analysis using a standard linear random effects model. This model made it possible to estimate the effects of legal origin on CSR scores by exploiting both within (over time) and between (across firms) variation in the

sample. An advantage of random effects models is that they allow us to include (and estimate the effect of) time invariant variables, among which is our key regressor of interest, i.e., legal origin. In the fixed effects model these time-invariant variables are absorbed by the intercept.

The baseline model we estimated was the following:

$$CSR_{it} = \alpha + \beta\ Legal\_Origins_i + \gamma_1 \ln(Total\_Assets)_{it} + \gamma_2 GDP_{it} + \gamma_3 G/GDP_{it} + \sum \delta_j Dyear_j + \sum \omega_k Dindustry_k + \sum \zeta_l X_{lt} + v_i + \varepsilon_{it}, \quad (3)$$

where for firm $i$ at time $t$ the dependent variable *CSR* was, in some specifications, the overall or the domain-specific CSR score while, in alternative ones, was the firm $i$'s deviation from the industry-average (overall or domain-specific) CSR score calculated at time $t$. *Legal_Origins* was our main variable of interest which, according to the implemented specification, took the form of a (0/1) dummy for countries belonging to the civil law group (variable *Civil Law*) or, alternatively, of a set of (0/1) dummies for countries belonging to the French, Scandinavian, German, and English legal origin groups (variables *French*, *Scandinavian*, *German,* and *English*), with the latter used as omitted benchmark. Among controls, $\ln(Total\_Assets)$ is the natural logarithm of the value of assets in US$ owned by firm $i$ at time $t$ and is a proxy for firm size, *GDP* and *G/GDP* are, respectively, the country per capita GDP in US$PPP and the country government expenditure (total expense and the net acquisition of nonfinancial assets) as percentage of GDP for firm $i$ at time $t$ (*GDP* has been scaled by 1000); *Dyear* and *Dindustry* are, respectively, dummy variables for each year with no missing observations in all the CSR criteria (2003 is the omitted benchmark) and dummies for the industry firm $i$ belongs to (the aerospace industry is the omitted category). In alternative specifications we also introduced a set of additional *X* controls at firm, industry, and country level which will be discussed in detail in Section 5.2 (*Robustness checks*). Finally, $\varepsilon_{it}$ is an idiosyncratic error, while $v_i$ captures a firm's time invariant characteristics. In all estimates, errors were clustered at country level to account for correlation of $v_i$ within countries.

In Table 5, Table 6 we report estimated findings for determinants of the overall CSR and of all CSR domains using the *Civil Law* dummy (Table 5) or the *French*, *Scandinavian*, *German* dummies (Table 6) as legal origin variables. The first specification confirmed previous descriptive findings, since common law countries performed better in the overall CSR score than civil law countries (Table 5, column 1). When considering the specific domains, the same holds true for community involvement and corporate governance (Table 5, columns 5 and 6). When considering the legal origin dummies, we found that the French legal origin was associated with higher CSR scores in the human resources domain than the English legal origin (Table 6, column 2), whereas the latter outperformed under the corporate governance and community involvement criteria (Table 6, columns 5 and 6). In terms of economic significance, the common law effect on corporate governance is remarkable (1.49 times the standard deviation of the dependent variable) and much stronger than the effect of the same legal origin group on community involvement (0.37 percent of the standard deviation) and the French law effect on human resources (0.37 percent of the standard deviation). Looking at the impact of our controls, we did not find evidence of substitutability between public welfare and corporate social responsibility (no evidence of negative and significant effect, while in very few cases a positive and significant effect).

In order to net out the effect of industry specific weights, we evaluated the robustness of the previous findings with the alternative approach of considering as the dependent variable deviations of CSR scores from industry averages for each company and CSR criterion. Regression results were again consistent with descriptive evidence (Table 7, Table 8), with common (civil) law firms above (below) their industry average when considering the overall score and the corporate governance and community involvement domains (Table 7, columns 5 and 6). In these estimates, firms belonging to the French legal origin family were more likely to be above their industry average score in the human resources domain (Table 8, column 2) than those in the English family. On the contrary, the latter tended to outperform all the other legal origin groups in the corporate governance and community involvement domains, as well as when considering the overall CSR score (Table 8, columns 5 and 6). Results are robust to the exclusion of industry dummies (see Tables A1 and A2 in Appendix A).

**Table 5.** Determinants of CSR over time: random effects.

| Variables | (1) Overall Score | (2) Human Resources | (3) Environment | (4) Business Behavior | (5) Corporate Governance | (6) Community Involvement | (7) Human Rights |
|---|---|---|---|---|---|---|---|
| ln(Total Assets) | −0.0163 | −0.0122 | 0.0768 | 0.0129 | −0.0310 | 0.0947 | 0.000988 |
|  | (0.0548) | (0.0661) | (0.105) | (0.0556) | (0.0931) | (0.126) | (0.0680) |
| Civil Law | −5.837 *** | 1.417 | −2.372 | −2.131 * | −25.36 *** | −6.922 *** | −1.061 |
|  | (1.378) | (1.816) | (1.882) | (1.174) | (2.818) | (1.500) | (1.069) |
| GDP | 0.0213 | 0.0305 | −0.166 | 0.00896 | 0.168 | −0.0515 | 0.0538 |
|  | (0.0900) | (0.0983) | (0.104) | (0.0764) | (0.146) | (0.110) | (0.0840) |
| G/GDP | 0.174 ** | 0.183 | 0.0875 | 0.129 | 0.173 | 0.301 * | 0.257 ** |
|  | (0.0848) | (0.154) | (0.108) | (0.0877) | (0.134) | (0.156) | (0.104) |
| Industry dummies | YES | YES | YES | YES | YES | YES | YES |
| Year dummies | YES | YES | YES | YES | YES | YES | YES |
| Observations | 6757 | 6757 | 6757 | 6757 | 6757 | 6757 | 6757 |
| Number of Countries | 27 | 27 | 27 | 27 | 27 | 27 | 27 |
| Number of Firms | 1834 | 1834 | 1834 | 1834 | 1834 | 1834 | 1834 |

Robust standard errors clustered at country level are reported in parentheses. *** $p < 0.01$, ** $p < 0.05$, * $p < 0.1$. Omitted categories: Aerospace (Industry); 2003 (Year).

**Table 6.** Determinants of CSR over time: random effects.

| Variables | (1) Overall Score | (2) Human Resources | (3) Environment | (4) Business Behavior | (5) Corporate Governance | (6) Community Involvement | (7) Human Rights |
|---|---|---|---|---|---|---|---|
| ln(Total Assets) | −0.00702 | 0.0111 | 0.0769 | 0.0204 | −0.0205 | 0.0994 | 0.00765 |
|  | (0.0582) | (0.0621) | (0.109) | (0.0573) | (0.0901) | (0.137) | (0.0674) |
| French Origins | −3.701 * | 6.595 ** | −2.048 | −0.254 | −23.50 *** | −4.299 * | 0.511 |
|  | (2.113) | (2.754) | (3.062) | (1.863) | (3.032) | (2.408) | (1.947) |
| Scandinavian Origins | −6.704 ** | −0.381 | −3.614 | −2.829 | −23.42 *** | −14.24 *** | −1.362 |
|  | (2.631) | (2.682) | (3.305) | (2.373) | (3.839) | (2.678) | (2.719) |
| German Origins | −7.015 *** | −1.399 | −2.303 | −3.108 ** | −26.87 *** | −7.034 *** | −1.942 * |
|  | (1.301) | (1.730) | (1.859) | (1.236) | (3.086) | (1.201) | (1.125) |
| GDP | 0.0336 | 0.0586 | −0.150 | 0.0193 | 0.141 | 0.0565 | 0.0584 |
|  | (0.0924) | (0.108) | (0.107) | (0.0791) | (0.146) | (0.114) | (0.0852) |
| G/GDP | 0.154 * | 0.125 | 0.0956 | 0.105 | 0.119 | 0.348 ** | 0.234 ** |
|  | (0.0793) | (0.115) | (0.108) | (0.0848) | (0.132) | (0.146) | (0.0927) |
| Industry dummies | YES | YES | YES | YES | YES | YES | YES |
| Year dummies | YES | YES | YES | YES | YES | YES | YES |
| Observations | 6757 | 6757 | 6757 | 6757 | 6757 | 6757 | 6757 |
| Number of Countries | 27 | 27 | 27 | 27 | 27 | 27 | 27 |
| Number of Firms | 1834 | 1834 | 1834 | 1834 | 1834 | 1834 | 1834 |

Robust standard errors clustered at country level are reported in parentheses. *** $p < 0.01$, ** $p < 0.05$, * $p < 0.1$. Omitted categories: Aerospace (Industry); 2003 (Year); English Origins.

**Table 7.** Determinants of deviations from industry average CSR over time: random effects.

| Variables | (1) Overall Score | (2) Human Resources | (3) Environ-ment | (4) Business Behavior | (5) Corporate Governance | (6) Community Involvement | (7) Human Rights |
|---|---|---|---|---|---|---|---|
| ln(Total Assets) | 0.0123 | 0.0307 | 0.0170 | 0.0637 | 0.0259 | 0.0664 | 0.0127 |
| | (0.0480) | (0.0689) | (0.0932) | (0.0493) | (0.0795) | (0.108) | (0.0558) |
| Civil Law | −5.335 *** | 1.107 | −2.768 * | −1.769 | −22.57 *** | −5.741 *** | −0.986 |
| | (1.250) | (1.556) | (1.534) | (1.085) | (2.870) | (1.094) | (0.969) |
| GDP | −0.0310 | 0.0156 | −0.152 * | 0.0102 | 0.0939 | −0.164 * | 0.0162 |
| | (0.0832) | (0.0925) | (0.0910) | (0.0821) | (0.136) | (0.0930) | (0.0737) |
| G/GDP | 0.118 | 0.165 | 0.163 * | 0.106 | 0.0636 | 0.107 | 0.211 ** |
| | (0.0866) | (0.146) | (0.0901) | (0.0818) | (0.114) | (0.114) | (0.0906) |
| Industry dummies | YES | YES | YES | YES | YES | YES | YES |
| Year dummies | YES | YES | YES | YES | YES | YES | YES |
| Observations | 6757 | 6757 | 6757 | 6757 | 6757 | 6757 | 6757 |
| Number of Countries | 27 | 27 | 27 | 27 | 27 | 27 | 27 |
| Number of Firms | 1834 | 1834 | 1834 | 1834 | 1834 | 1834 | 1834 |

Robust standard errors clustered at country level are reported in parentheses. *** $p < 0.01$, ** $p < 0.05$, * $p < 0.1$. Omitted categories: Aerospace (Industry); 2003 (Year).

**Table 8.** Determinants of deviations from industry average CSR over time: random effects.

| Variables | (1) Overall Score | (2) Human Resources | (3) Environment | (4) Business Behavior | (5) Corporate Governance | (6) Community Involvement | (7) Human Rights |
|---|---|---|---|---|---|---|---|
| ln(Total Assets) | 0.0207 | 0.0487 | 0.0152 | 0.0695 | 0.0372 | 0.0720 | 0.0194 |
| | (0.0494) | (0.0609) | (0.0951) | (0.0494) | (0.0784) | (0.117) | (0.0549) |
| French Origins | −3.445 * | 5.436 ** | −2.843 | −0.227 | −20.46 *** | −3.378 * | 0.484 |
| | (1.880) | (2.363) | (2.562) | (1.652) | (2.755) | (1.934) | (1.694) |
| Scandinavian Origins | −5.753 ** | −0.333 | −4.109 | −2.578 | −20.33 *** | −11.16 *** | −0.566 |
| | (2.266) | (2.485) | (2.868) | (2.154) | (3.272) | (1.959) | (2.353) |
| German Origins | −6.408 *** | −1.199 | −2.442 * | −2.522 ** | −24.23 *** | −6.100 *** | −1.927 ** |
| | (1.193) | (1.492) | (1.429) | (1.131) | (3.216) | (0.891) | (0.971) |
| GDP | −0.0244 | 0.0383 | −0.133 | 0.0221 | 0.0611 | −0.0810 | 0.0100 |
| | (0.0842) | (0.0953) | (0.0970) | (0.0829) | (0.132) | (0.0846) | (0.0726) |
| G/GDP | 0.0935 | 0.110 | 0.180** | 0.0894 | −0.00394 | 0.137 | 0.178** |
| | (0.0763) | (0.112) | (0.0907) | (0.0773) | (0.106) | (0.0902) | (0.0803) |
| Industry dummies | YES | YES | YES | YES | YES | YES | YES |
| Year dummies | YES | YES | YES | YES | YES | YES | YES |
| Observations | 6757 | 6757 | 6757 | 6757 | 6757 | 6757 | 6757 |
| Number of Countries | 27 | 27 | 27 | 27 | 27 | 27 | 27 |
| Number of Firms | 1834 | 1834 | 1834 | 1834 | 1834 | 1834 | 1834 |

Robust standard errors clustered at country level are reported in parentheses. *** $p < 0.01$, ** $p < 0.05$, * $p < 0.1$. Omitted categories: Aerospace (Industry); 2003 (Year); English Origins.

We repeated our estimates for any single sustainability variable in order to verify which of them was driving the aggregate domain results (see Table A5 in Appendix A). What we observed here is that the superior performance of the civil law origin was significant for all items for which we had a sufficient number of observations in the corporate governance and community involvement domains. Results were mixed in the human rights domain (where the civil law origin significantly outperformed in the promotion of labor relations and encouraging employee participation sustainability drivers, while it significantly underperformed in the improvement of health and safety conditions and respect and management of working hours sustainability drivers) and in the human resources domain (where the civil law origin significantly outperformed in the respect for freedom of association and right to collective bargaining and non-discrimination sustainability drivers). Note, however, that when looking at the performance of families within the two legal origin groups, the French family outperformed in six of the human resources sustainability drivers (promotion of labor relations, encouraging employee participation, training and development, responsible management and restructurings, career management, and promotion of employability), thereby confirming our previous findings on this point.

Statistical tests and regressions highlighted lack of a statistically significant impact of legal origins on CSR scores regarding the environment criteria, both in aggregate domains (column 3 in Tables 5–8) and in the sustainability driver estimates (Table A5 in Appendix A), with the exception of the management of atmospheric emissions. A plausible explanation for this hinges on CSR convergence between firms in civil and common law countries in those specific domains, convergence which we already envisaged in the inspection of CSR score time dynamics (Figure 1). This finding is consistent with Meyer's [42] prediction that regulatory processes would generate more standardized and rationalized concerns on environmental issues. More specifically, Meyer [42] argued that similarities in terms of environmental concerns in Western countries may be a rationale for convergence in CSR policies on environment, However, this may not be the case for CSR activities related to labor issues as labor markets, are still deregulated across liberal market economies. Our evidence on the lack of CSR convergence between civil and common law countries in terms of human resources confirms the latter prediction.

In order to test for convergence in the last ten-year sample period, we averaged each domain-specific CSR score over five main periods (2003–2005, 2006–2007, 2008–2009, 2010–2011, 2012–2013), constructed for each domain the growth rate of CSR between the first and the last period, and regressed it on the same controls as in Equation (3), also adding the level of CSR score for the relevant domain in the first period (2003–2005). CSR growth rate is calculated as $\frac{CSR_{i,t=2012-13} - CSR_{i,t=2003-05}}{CSR_{i,t=2003-05}}$. A negative and significant coefficient of the first-period level would support the convergence hypothesis.

Results from Ordinary-Least-Squares (OLS) regressions are reported in Table 9, Table 10 and confirm the convergence hypothesis, since the coefficient of the first-period level variable (*Score Level 03–05*) was negative and significant for all CSR domains—hence indicating a general convergence for all criteria—while the coefficients of the legal origin variables were not (or weakly) significant just for the business behavior and environment domains (Table 8, Table 9, columns 3 and 4). Note that the presence of significant convergence effects also in the other CSR domains does not contradict descriptive evidence in Figure 1. What is probably at work is a convergence process which is mostly within, but not between, legal origin groups in these cases. This justifies both convergence and significant coefficients of the legal origin dummies. Evidence of within legal group convergence is omitted for reasons of space and is available from the authors upon request.

**Table 9.** CSR convergence between civil and common law countries.

| Variables | (1) Overall Score | (2) Human Resources | (3) Environment | (4) Business Behavior | (5) Corporate Governance | (6) Community Involvement | (7) Human Rights |
|---|---|---|---|---|---|---|---|
| ln(Total Assets) | −0.00508 | 0.00206 | 0.0168 | −0.00574 | −0.0121 * | 0.0583 | 0.000891 |
| | (0.00539) | (0.00786) | (0.0294) | (0.00704) | (0.00690) | (0.0461) | (0.00589) |
| Civil Law | 0.0513 | 0.236 *** | −0.0781 | 0.0891 * | −0.472 *** | 0.221 * | 0.179 *** |
| | (0.0399) | (0.0293) | (0.193) | (0.0454) | (0.108) | (0.119) | (0.0342) |
| GDP | −0.00116 | 0.000704 | 0.00906 | −0.00543 | −0.00173 | −0.00872 | −0.00592 * |
| | (0.00443) | (0.00391) | (0.0121) | (0.00351) | (0.00843) | (0.00755) | (0.00284) |
| G/GDP | 0.00260 | 0.00863 ** | 0.00227 | −0.00422 | 9.64e−05 | −0.0311 * | −0.00124 |
| | (0.00261) | (0.00326) | (0.00730) | (0.00289) | (0.00497) | (0.0153) | (0.00197) |
| Score Level (03–05) | −0.0267 *** | −0.0208 *** | −0.0443 *** | −0.0262 *** | −0.0297 *** | −0.0359 *** | −0.0165 *** |
| | (0.00136) | (0.00265) | (0.00711) | (0.00384) | (0.00737) | (0.0118) | (0.000905) |
| Dummy 2006–2007 | 0.0606 * | 0.0142 | −0.165 | −0.00817 | −0.0471 | −0.144 | 0.0865 ** |
| | (0.0314) | (0.0293) | (0.211) | (0.0315) | (0.0348) | (0.166) | (0.0315) |
| Dummy 2008–2009 | 0.0411 | −0.0227 | −0.199 | 0.00918 | −0.0428 | −0.0524 | 0.0936 ** |
| | (0.0329) | (0.0311) | (0.236) | (0.0401) | (0.0477) | (0.150) | (0.0360) |
| Dummy 2010–2011 | 0.0371 | −0.0349 | −0.219 | 0.0128 | −0.0413 | −0.0490 | 0.0963 ** |
| | (0.0370) | (0.0360) | (0.249) | (0.0445) | (0.0578) | (0.173) | (0.0379) |
| Dummy 2012–2013 | 0.0422 | −0.0317 | −0.232 | 0.0190 | −0.0425 | −0.0826 | 0.103 ** |
| | (0.0425) | (0.0381) | (0.256) | (0.0472) | (0.0621) | (0.191) | (0.0407) |
| Industry dummies | YES | YES | YES | YES | YES | YES | YES |
| Observations | 1231 | 1245 | 1237 | 1245 | 1245 | 1240 | 1231 |
| R−squared | 0.728 | 0.549 | 0.452 | 0.570 | 0.436 | 0.272 | 0.552 |

Robust standard errors clustered at country level are reported in parentheses, *** $p < 0.01$, ** $p < 0.05$, * $p < 0.1$. Dependent variable: $\frac{CSR_{i,t=2012-13} - CSR_{i,t=2003-2005}}{CSR_{i,t=2003-2005}}$. Omitted categories: Aerospace (Industry); 2003–2005 (Year).

**Table 10.** CSR convergence between countries with different legal origins.

| Variables | (1) Overall Score | (2) Human Resources | (3) Environment | (4) Business Behavior | (5) Corporate Governance | (6) Community Involvement | (7) Human Rights |
|---|---|---|---|---|---|---|---|
| ln(Total Assets) | −0.00743 | −0.000194 | 0.0142 | −0.00589 | −0.0124 | 0.0570 | 0.000811 |
| | (0.00493) | (0.00795) | (0.0296) | (0.00694) | (0.00749) | (0.0474) | (0.00602) |
| French Origins | 0.125 *** | 0.304 *** | −0.00773 | 0.0919 | −0.464 *** | 0.235 ** | 0.187 *** |
| | (0.0430) | (0.0388) | (0.215) | (0.0638) | (0.105) | (0.106) | (0.0463) |
| Scandinavian Origins | −0.0599 | 0.0982 *** | −0.301 | 0.0645 | −0.534 *** | −0.139 | 0.228 *** |
| | (0.0541) | (0.0330) | (0.225) | (0.0721) | (0.141) | (0.141) | (0.0535) |
| German Origins | −0.00368 | 0.193 *** | −0.102 | 0.0908 * | −0.465 *** | 0.268 | 0.162 *** |
| | (0.0386) | (0.0192) | (0.179) | (0.0491) | (0.129) | (0.261) | (0.0306) |
| GDP | 0.00189 | 0.00417 | 0.0140 | −0.00494 | −0.000479 | −0.00199 | −0.00669 ** |
| | (0.00303) | (0.00265) | (0.0147) | (0.00373) | (0.00812) | (0.00915) | (0.00301) |
| G/GDP | $2.17 \times 10^{-05}$ | 0.00711 ** | 0.00255 | −0.00391 | 0.000981 | −0.0249 ** | −0.00275 |
| | (0.00248) | (0.00253) | (0.00753) | (0.00369) | (0.00687) | (0.0116) | (0.00311) |
| Score Level (03–05) | −0.0265 *** | −0.0210 *** | −0.0436 *** | −0.0262 *** | −0.0297 *** | −0.0363 *** | −0.0165 *** |
| | (0.00140) | (0.00262) | (0.00704) | (0.00382) | (0.00731) | (0.0117) | (0.000924) |
| Dummy 2006–2007 | 0.0451 | −0.000488 | −0.190 | −0.0100 | −0.0518 | −0.166 | 0.0886 ** |
| | (0.0276) | (0.0256) | (0.216) | (0.0320) | (0.0350) | (0.163) | (0.0314) |
| Dummy 2008–2009 | 0.0341 | −0.0328 | −0.226 | 0.00604 | −0.0511 | −0.0998 | 0.101 ** |
| | (0.0303) | (0.0261) | (0.246) | (0.0397) | (0.0465) | (0.165) | (0.0393) |
| Dummy 2010–2011 | 0.0298 | −0.0465 | −0.251 | 0.00907 | −0.0514 | −0.106 | 0.106 ** |
| | (0.0328) | (0.0292) | (0.262) | (0.0441) | (0.0570) | (0.192) | (0.0424) |
| Dummy 2012–2013 | 0.0281 | −0.0499 | −0.271 | 0.0147 | −0.0538 | −0.145 | 0.112 ** |
| | (0.0358) | (0.0308) | (0.271) | (0.0459) | (0.0599) | (0.208) | (0.0443) |
| | 0.0451 | −0.000488 | −0.190 | −0.0100 | −0.0518 | −0.166 | 0.0886 ** |
| Industry dummies | YES | YES | YES | YES | YES | YES | YES |
| Observations | 1231 | 1245 | 1237 | 1245 | 1245 | 1240 | 1231 |
| R-squared | 0.739 | 0.564 | 0.455 | 0.570 | 0.437 | 0.278 | 0.553 |

Robust standard errors clustered at country level are reported in parentheses, *** $p < 0.01$, ** $p < 0.05$, * $p < 0.1$. Dependent variable: $\frac{CSR_{i,t=2012-13} - CSR_{i,t=2003-2005}}{CSR_{i,t=2003-2005}}$. Omitted categories: Aerospace (Industry); 2003–2005 (Year); English Origins.

The observed convergence effect may be in part due to the fact that globalization reduced the influence of legal origins on corporate practices. Another plausible explanation for the convergence in the environment domain is the generalized adoption of world standards (i.e., the Forest Stewardship Council standard on the use of sustainable paper, which developed quite rapidly around the world). A third and related rationale is that companies have increasingly adopted benchmarking practices in their competitive strategies. The application of these to environmental standards may contribute to explaining both the reinforcement of global social norms on environmental sustainability and the convergence to common industry standards. All these potential explanations are not the main core of this paper and may indeed be a promising ground for further studies on CSR.

To conclude this section, our hypotheses found empirical support also under the econometric analysis, taking into account industry heterogeneity and time structure of our dataset. Companies in common law countries performed better under community involvement and corporate governance criteria, while firms in countries with French legal origins received higher ratings in the human resources domain. Finally, under the environment criteria there was no significant difference among countries in terms of their legal origins, since in this domain they tended to converge more significantly than in the others to a common global industry standard.

### 5.2. Robustness Checks

A problem that usually arises when running the standard linear random effects model concerns the assumption of zero correlation between the firm characteristics $v_i$ and all the other regressors. If this assumption can be realistic with respect to the legal origin variables, it may be posed under discussion when considering the other regressors. We could not solve the problem with a fixed effect model, since the effect of the (time-invariant) main variable of interest (*Legal_Origin*) would be absorbed in firm-specific intercepts. We therefore ran our robustness check by implementing Mundlak's [43] approach. The latter implies the re-estimation of the random effect model with the addition of group-means of the time variant variables *GDP*, *G/GDP*, and *Total Assets*, which we named, respectively, $\overline{GDP}$, $\overline{G/GDP}$ and $\overline{Total\ Assets}$. All the results were consistent with those commented in the previous sections and are reported in Table 11, Table 12, where the dependent variable was the firms' CSR score (overall and in the specific sustainability drivers) and in Table 13, Table 14, where the dependent variable was firms' deviation from their CSR industry average (overall and in the different sub-domains). For random effect estimations with Mundlak correction without industry dummies see Tables A3 and A4 in Appendix A.

Another potential bias in our estimates (arising from the sample composition of the VIGEO data) derived from non-random attrition, since the probability that firms entered and exited our panel may have depended on observable and/or unobservable factors possibly correlated with the main variable of interest (the CSR scores). In order to reduce this potential bias in the main estimates, we first estimated the firms' attrition probability, controlling for year, sector, and country effects with the addition of the country per-capita GDP and a proxy for the difficulty of doing business in a given country (i.e., the number of procedures necessary to start up a new business). These data vary at yearly basis and were taken from the "Doing Business" panel available at http://www.doingbusiness.org/custom-query--variable: *Starting a Business (Procedures Numbers)*.

We then used the predicted attrition probabilities to (inversely) weight each observation. The weights were constructed as *1/p(Ai)*, where *p(Ai)* is the estimated probability of attrition for each firm. With such a weighting method, each observation in the main equation is inversely weighted by its attrition probability so that less importance in the estimation is given to those firms more likely to attrite.

Estimation results of the attrition probit model and of the main CSR equations through pooled OLS and weighted least squares (WLS) are reported, respectively, in column A and columns 1–14 of Table 15. Since, in general, WLS estimates did not significantly differ from the pooled OLS ones and were consistent with those reported in Table 5, Table 6, we can conclude that firms' non-random attrition was not likely to be the main driver of our results.

**Table 11.** Determinants of CSR over time: random effects (Mundlak correction).

| Variables | (1) Overall Score | (2) Human Resources | (3) Environment | (4) Business Behavior | (5) Corporate Governance | (6) Community Involvement | (7) Human Rights |
|---|---|---|---|---|---|---|---|
| ln(Total Assets) | −0.253 | −0.186 | 0.0477 | 0.00778 | 0.158 | −0.404 | −0.673 |
| | (0.378) | (0.640) | (0.612) | (0.584) | (0.499) | (0.861) | (0.441) |
| Civil Law | −6.569 *** | −0.124 | −3.573 ** | −2.753 *** | −25.82 *** | −6.569 *** | −1.437 |
| | (1.265) | (1.535) | (1.720) | (1.044) | (2.944) | (1.277) | (1.018) |
| GDP | 0.521 ** | 0.802 *** | 0.376 | 0.703 *** | 0.517 * | 0.783 ** | 0.704 *** |
| | (0.252) | (0.252) | (0.298) | (0.237) | (0.308) | (0.355) | (0.270) |
| G/GDP | 0.171 * | 0.0399 | 0.0374 | 0.104 | 0.138 | 0.525 * | 0.292 ** |
| | (0.0999) | (0.0968) | (0.120) | (0.113) | (0.208) | (0.282) | (0.117) |
| $\overline{\text{Total Assets}}$ | 0.277 | 0.209 | 0.0490 | 0.0203 | −0.195 | 0.538 | 0.731 |
| | (0.402) | (0.673) | (0.641) | (0.626) | (0.496) | (0.870) | (0.494) |
| $\overline{\text{GDP}}$ | −0.621 ** | −0.870 *** | −0.664 * | −0.790 *** | −0.401 | −0.978 *** | −0.756 *** |
| | (0.246) | (0.250) | (0.340) | (0.252) | (0.249) | (0.334) | (0.267) |
| $\overline{\text{G/GDP}}$ | 0.0214 | 0.271 | 0.129 | 0.0332 | 0.0576 | −0.365 | −0.0583 |
| | (0.149) | (0.178) | (0.171) | (0.141) | (0.280) | (0.286) | (0.132) |
| Industry dummies | YES | YES | YES | YES | YES | YES | YES |
| Year dummies | YES | YES | YES | YES | YES | YES | YES |
| Observations | 6757 | 6757 | 6757 | 6757 | 6757 | 6757 | 6757 |
| Number of Countries | 27 | 27 | 27 | 27 | 27 | 27 | 27 |
| Number of Firms | 1834 | 134 | 1834 | 1834 | 1834 | 1834 | 1834 |

Robust standard errors clustered at country level are reported in parentheses. *** $p < 0.01$, ** $p < 0.05$, * $p <0.1$. Omitted categories: Aerospace (Industry); 2003 (Year).

**Table 12.** Determinants of CSR over time: random effects (Mundlak correction).

| Variables | (1) Overall Score | (2) Human Resources | (3) Environment | (4) Business Behavior | (5) Corporate Governance | (6) Community Involvement | (7) Human Rights |
|---|---|---|---|---|---|---|---|
| ln(Total Assets) | −0.277 | −0.263 | 0.0561 | −0.0194 | 0.126 | −0.444 | −0.697 |
| | (0.386) | (0.663) | (0.64) | (0.589) | (0.520) | (0.863) | (0.453) |
| French Origins | −4.876 ** | 4.421 ** | −4.138 | −1.472 | −24.17 *** | −4.532 ** | −0.312 |
| | (1.981) | (2.204) | (2.855) | (1.715) | (3.193) | (2.230) | (1.851) |
| Scandinavian Origins | −6.365 *** | −1.281 | −4.204 | −2.381 | −23.09 *** | −12.37 *** | −0.482 |
| | (2.461) | (2.784) | (3.139) | (2.223) | (3.731) | (2.419) | (2.766) |
| German Origins | −7.472 *** | −2.305 | −3.193 ** | −3.463 *** | −27.05*** | −6.841 *** | −2.145 ** |
| | (1.315) | (1.567) | (1.594) | (1.094) | (3.267) | (1.088) | (1.030) |
| GDP | 0.494 ** | 0.715 *** | 0.385 | 0.674 *** | 0.485 * | 0.729 ** | 0.681 ** |
| | (0.248) | (0.220) | (0.303) | (0.233) | (0.287) | (0.357) | (0.266) |
| G/GDP | 0.160 * | 0.00280 | 0.0418 | 0.0909 | 0.120 | 0.510 * | 0.280 ** |
| | (0.0969) | (0.0871) | (0.120) | (0.111) | (0.202) | (0.281) | (0.113) |
| $\overline{\text{Total Assets}}$ | 0.311 | 0.309 | 0.0366 | 0.0542 | −0.151 | 0.582 | 0.761 |
| | (0.413) | (0.703) | (0.645) | (0.635) | (0.520) | (0.871) | (0.510) |
| $\overline{\text{GDP}}$ | −0.591 ** | −0.746 *** | −0.665 * | −0.764 *** | −0.412 * | −0.814 ** | −0.746 *** |
| | (0.250) | (0.222) | (0.354) | (0.258) | (0.241) | (0.346) | (0.263) |
| $\overline{\text{G/GDP}}$ | −0.00297 | 0.242 * | 0.147 | 0.0144 | −0.00705 | −0.284 | −0.0861 |
| | (0.141) | (0.129) | (0.167) | (0.141) | (0.285) | (0.261) | (0.122) |
| Industry dummies | YES | YES | YES | YES | YES | YES | YES |
| Year dummies | YES | YES | YES | YES | YES | YES | YES |
| Observations | 6757 | 6757 | 6757 | 6757 | 6757 | 6757 | 6757 |
| Number of Countries | 27 | 27 | 27 | 27 | 27 | 27 | 27 |
| Number of Firms | 1834 | 1834 | 1834 | 1834 | 1834 | 1834 | 1834 |

Robust standard errors clustered at country level are reported in parentheses. *** $p < 0.01$, ** $p < 0.05$, * $p < 0.1$. Omitted categories: Aerospace (Industry); 2003 (Year); English Origins.

**Table 13.** Determinants of deviations from industry average CSR: random effects (Mundlak correction).

| Variables | (1) Overall Score | (2) Human Resources | (3) Environment | (4) Business Behavior | (5) Corporate Governance | (6) Community Involvement | (7) Human Rights |
|---|---|---|---|---|---|---|---|
| ln(Total Assets) | −0.0264 | 0.339 | −0.248 | 0.463 | 0.518 | −0.118 | −0.544 |
| | (0.274) | (0.480) | (0.408) | (0.423) | (0.542) | (0.516) | (0.537) |
| Civil Law | −6.020 *** | −0.0562 | −3.404 ** | −2.364 ** | −23.30 *** | −5.899 *** | −1.438 |
| | (1.186) | (1.317) | (1.599) | (0.982) | (3.011) | (1.069) | (0.911) |
| GDP | 0.410 | 0.692 *** | 0.372 | 0.600 ** | 0.295 | 0.390 | 0.613 *** |
| | (0.269) | (0.264) | (0.313) | (0.249) | (0.324) | (0.341) | (0.218) |
| G/GDP | 0.0736 | 0.0205 | 0.144 | 0.0621 | −0.0908 | 0.151 | 0.199 ** |
| | (0.0880) | (0.104) | (0.116) | (0.0988) | (0.115) | (0.124) | (0.0897) |
| $\overline{\text{Total Assets}}$ | 0.0537 | −0.309 | 0.298 | −0.409 | −0.511 | 0.204 | 0.597 |
| | (0.297) | (0.503) | (0.440) | (0.458) | (0.538) | (0.529) | (0.579) |
| $\overline{\text{GDP}}$ | −0.507 ** | −0.745 *** | −0.615 * | −0.664 *** | −0.202 | −0.619 * | −0.670 *** |
| | (0.254) | (0.250) | (0.338) | (0.242) | (0.264) | (0.321) | (0.210) |
| $\overline{\text{G/GDP}}$ | 0.0868 | 0.239 | 0.0389 | 0.0634 | 0.239 | −0.0741 | 0.0144 |
| | (0.127) | (0.160) | (0.163) | (0.126) | (0.219) | (0.131) | (0.0984) |
| Industry dummies | YES | YES | YES | YES | YES | YES | YES |
| Year dummies | YES | YES | YES | YES | YES | YES | YES |
| Observations | 6757 | 6757 | 6757 | 6757 | 6757 | 6757 | 6757 |
| Number of Countries | 27 | 27 | 27 | 27 | 27 | 27 | 27 |
| Number of Firms | 1834 | 1834 | 1834 | 1834 | 1834 | 1834 | 1834 |

Robust standard errors clustered at country level are reported in parentheses. *** $p < 0.01$, ** $p < 0.05$, * $p < 0.1$. Omitted categories: Aerospace (Industry); 2003 (Year).

**Table 14.** Determinants of deviations from industry average CSR: random effects (Mundlak correction).

| Variables | (1) Overall Score | (2) Human Resources | (3) Environment | (4) Business Behavior | (5) Corporate Governance | (6) Community Involvement | (7) Human Rights |
|---|---|---|---|---|---|---|---|
| ln(Total Assets) | −0.0534 | 0.266 | −0.234 | 0.442 | 0.474 | −0.161 | −0.569 |
| | (0.285) | (0.509) | (0.412) | (0.429) | (0.553) | (0.514) | (0.550) |
| French Origins | −4.528 *** | 3.706 ** | −4.142 | −1.392 | −21.34 *** | −3.969 ** | −0.379 |
| | (1.752) | (1.815) | (2.527) | (1.521) | (2.934) | (1.861) | (1.573) |
| Scandinavian Origins | −5.843 *** | −0.985 | −4.071 | −2.367 | −21.02 *** | −10.82 *** | −0.0674 |
| | (2.138) | (2.438) | (2.739) | (1.997) | (3.478) | (1.888) | (2.341) |
| German Origins | −6.813 *** | −1.865 | −2.932 ** | −2.865 *** | −24.61 *** | −6.244 *** | −2.166 ** |
| | (1.242) | (1.393) | (1.402) | (1.043) | (3.367) | (0.887) | (0.911) |
| GDP | 0.381 | 0.609 *** | 0.387 | 0.577 ** | 0.251 | 0.333 | 0.589 *** |
| | (0.261) | (0.219) | (0.315) | (0.246) | (0.304) | (0.334) | (0.205) |
| G/GDP | 0.0606 | −0.0159 | 0.151 | 0.0518 | −0.115 | 0.134 | 0.186 ** |
| | (0.0837) | (0.0969) | (0.115) | (0.0970) | (0.107) | (0.115) | (0.0856) |
| $\overline{\text{Total Assets}}$ | 0.0888 | −0.218 | 0.279 | −0.383 | −0.454 | 0.250 | 0.629 |
| | (0.312) | (0.540) | (0.447) | (0.466) | (0.547) | (0.527) | (0.596) |
| $\overline{\text{GDP}}$ | −0.476 * | −0.633 *** | −0.621 * | −0.638 ** | −0.194 | −0.468 | −0.668 *** |
| | (0.254) | (0.213) | (0.344) | (0.248) | (0.261) | (0.328) | (0.194) |
| $\overline{\text{G/GDP}}$ | 0.0676 | 0.218 * | 0.0584 | 0.0544 | 0.182 | −0.00436 | −0.0188 |
| | (0.121) | (0.122) | (0.156) | (0.127) | (0.229) | (0.0964) | (0.0910) |
| Industry dummies | YES | YES | YES | YES | YES | YES | YES |
| Year dummies | YES | YES | YES | YES | YES | YES | YES |
| Observations | 6757 | 6757 | 6757 | 6757 | 6757 | 6757 | 6757 |
| Number of Countries | 27 | 27 | 27 | 27 | 27 | 27 | 27 |
| Number of Firms | 1834 | 1834 | 1834 | 1834 | 1834 | 1834 | 1834 |

Robust standard errors clustered at country level are reported in parentheses. *** $p < 0.01$, ** $p < 0.05$, * $p < 0.1$. Omitted categories: Aerospace (Industry); 2003 (Year); English Origins.

**Table 15.** Determinants of CSR: correction for attrition bias.

| Model: | (A) PROBIT | Model: | (1) OLS | (2) WLS | (3) OLS | (4) WLS | (5) OLS | (6) WLS | (7) OLS | (8) WLS | (9) OLS | (10) WLS | (11) OLS | (12) WLS | (13) OLS | (14) WLS |
|---|---|---|---|---|---|---|---|---|---|---|---|---|---|---|---|---|
| Dep. Var.: | Prob. attrition | Dep. Var.: | Overall Score | | Human Resources | | Environment | | Business Behavior | | Corporate Governance | | Community Involvement | | Human Rights | |
| | | | | | | | PANEL A | | | | | | | | | |
| GDP | −0.13 *** | French Or. | −4.48 * | −3.75 * | 5.64 ** | 7.12 *** | −4.10 | −3.05 | −0.88 | −0.19 | −23.7 *** | −23.7 *** | −3.7 | −2.60 | 0.19 | 1.02 |
| | (0.01) | | (2.245) | (2.08) | (2.45) | (2.33) | (3.08) | (2.89) | (1.88) | (1.6) | (2.98) | (2.65) | (2.46) | (2.23) | (2.17) | (2.2) |
| N. proc. to start a business | 0.09 *** | Scand. Or. | −8.29 *** | −9.07 *** | −2.91 | −3.26 | −7.27 * | −8.05 ** | −3.56 | −3.62 | −24 *** | −24.5 *** | −13.9 *** | −14.82 *** | −2.37 | −3.7 |
| | (0.02) | | (2.68) | (2.69) | (2.84) | (2.88) | (3.59) | (3.78) | (2.24) | (2.23) | (3.55) | (3.18) | (2.70) | (2.71) | (2.82) | (3.11) |
| | | German Or. | −6.15 *** | −5.9 *** | −0.01 | 1.00 | −2.18 | −1.95 | −2.62 ** | −2.63 ** | −25.4 *** | −25.4 *** | −6.18 *** | −5.734 *** | −0.96 | −0.52 |
| | | | (1.56) | (1.42) | (1.94) | (2.04) | (1.87) | (1.73) | (1.24) | (1.18) | (2.95) | (2.49) | (1.47) | (1.37) | (1.27) | (1.45) |
| Year d. | YES | Controls | YES | YES | YES | YES | YES | YES | YES | YES | YES | YES | YES | YES | YES | YES |
| Industry d. | YES | Year d. | YES | YES | YES | YES | YES | YES | YES | YES | YES | YES | YES | YES | YES | YES |
| Country d. | YES | Industry d. | YES | YES | YES | YES | YES | YES | YES | YES | YES | YES | YES | YES | YES | YES |
| Obs. | 18,673 | Obs | 5908 | 5888 | 5908 | 5888 | 5908 | 5888 | 5908 | 5888 | 5908 | 5888 | 5908 | 5888 | 5908 | 5888 |
| | | | | | | | PANEL B | | | | | | | | | |
| | | Civil Law | −5.76 *** | −5.40 *** | 1.66 | 2.756 | −3.22 | −2.77 | −2.11 * | −1.88 * | −24.72 *** | −24.72 *** | −5.94 *** | −5.345 *** | −0.68 | −0.23 |
| | | | (1.50) | (1.37) | (1.53) | (1.63) | (2.17) | (2.00) | (1.19) | (1.03) | (2.63) | (2.24) | (1.59) | (1.49) | (1.26) | (1.32) |
| | | Controls | YES | YES | YES | YES | YES | YES | YES | YES | YES | YES | YES | YES | YES | YES |
| | | Year d. | YES | YES | YES | YES | YES | YES | YES | YES | YES | YES | YES | YES | YES | YES |
| | | Industry d. | YES | YES | YES | YES | YES | YES | YES | YES | YES | YES | YES | YES | YES | YES |
| | | Obs. | 5908 | 5888 | 5908 | 5888 | 5908 | 5888 | 5908 | 5888 | 5908 | 5888 | 5908 | 5888 | 5908 | 5888 |

Notes: [1] model (A): robust standard errors clustered at firm level are reported in parentheses; [2] models (1–14): robust standard errors clustered at country level are reported in parentheses; the weights for the WLS models were calculated as $1/P(A_i)$, where $P(A_i)$ is the predicted attrition probability from the attrition prob. model in column (A); [3] *** $p < 0.01$, ** $p < 0.05$, * $p < 0.1$. [4] Omitted categories: Aerospace (Industry); 2004 (Year); English Origins (Panel A). [5] Controls: Total Assets, GDP, G/GDP; [6] OLS = Ordinary Least Squares (pooled); WLS = Weighted Least Squares.

Finally, in order to check whether our main findings were driven by VIGEO's industry classification, other macroeconomic country-specific characteristics, or firm/industry-specific features, we re-estimated our baseline specification in Table 6 by controlling for industry categories, firms' profitability (i.e., return on assets, ROA), and industrial concentration (proxied for by industry net-profit margins, NPM) from the COMPUSTAT dataset, as well as for the country inflation rate (*inflation_rate*) calculated on a yearly basis by the World Bank. As in Giroud and Mueller [44], industry net-profit margins have been calculated as the ratio between firms' "Operating Income Before Depreciation" and "Sales" averaged at industry level (data source: COMPUSTAT).

The new estimation results are shown in Table 16. Overall, despite the loss of several observations after merging the VIGEO and COMPUSTAT datasets, our core findings were still robust to the introduction of the new controls. Incidentally, high industrial concentration was associated with reduced CSR performance in all the domains but the human rights one. This last evidence only partially supports Campbell's [45] hypothesis that firms act in a less socially responsible way if there is either too much or too little competition. To better test the Campbell's hypothesis about a non-linear effect of sector competition on firm's CSR behaviour, in a different specification, we introduced the square of *NPM* and found that the latter is not significant (available upon request). Similar evidence against a non-linear relationship between CSR and competition is shown by Chih et al. [46].

A demand-based interpretation for the negative correlation between concentration and CSR-performance in the majority of CSR-domains is that when competition is low (and stakeholders face few alternatives) firms have fewer incentives to invest in "strategic philanthropy" to signal their competitive advantages (see, among others, Shleifer [47] and Porter and Kramer [48]). Conversely, a possible explanation for the positive impact of industrial concentration on CSR performance in the human rights domain may derive from a combination of several domain-specific factors counterbalancing the absence of alternatives for the consumers disappointed with firm's unethical behavior, i.e., for instance, the stricter international regulations protecting human rights relative to international standards in other domains, the higher visibility to the public opinion of human rights violations (with the consequent higher risk of a bad reputation for the deviating firm), the higher the sensitivity of the average consumer (even if less educated) to the firm's acts against human rights, etc. A relevant example on this point is the "Behind the brands" Oxfam's campaign on the 10 largest food industries. Oxfam developed social and environmental ratings for such companies and asked citizens to take actions by sending e-mails to them for approval/disapproval of their behaviour. After one year of campaign around 426,000 actions were taken and 9 out 10 companies announced the decision to improve their CSR practices. This is an example on how higher visibility to the public opinion may lead global companies with higher market power to have relatively more developed CSR policies (http://www.oxfam.org/en/grow/campaigns/behind-brands accessed at 27 of May 2014).

**Table 16.** Robustness checks.

| Variables | (1) | (2) | (3) | (4) | (5) | (6) | (7) | (8) | (9) | (10) | (11) | (12) | (13) | (14) |
|---|---|---|---|---|---|---|---|---|---|---|---|---|---|---|
| | *Overall Score* | | *Human Resources* | | *Environment* | | *Business Behavior* | | *Corporate Governance* | | *Community Involvement* | | *Human Rights* | |
| French Origins | 0.781 | −2.707 | 9.402 *** | 6.029 ** | 3.150 | −1.623 | 2.740 | −0.200 | −17.42 *** | −23.45 *** | 2.450 | −3.546 | 5.064 * | 0.964 |
| | (3.082) | (2.915) | (3.402) | (2.796) | (3.797) | (2.923) | (2.363) | (2.453) | (4.723) | (4.830) | (3.575) | (3.552) | (2.831) | (2.365) |
| Scandinavian Origins | −1.124 | −5.860 | 4.280 | −0.389 | 0.973 | −3.838 | 1.343 | −2.269 | −15.06 *** | −23.27 *** | −7.249 ** | −13.87 *** | 4.228 | −1.170 |
| | (3.277) | (3.696) | (2.751) | (3.229) | (3.983) | (3.557) | (2.543) | (2.831) | (5.142) | (5.776) | (3.465) | (4.331) | (2.995) | (3.079) |
| German Origins | −8.360 ** | −9.413 ** | −1.960 | −4.464 | −0.626 | −4.762 | −2.966 | −4.359 | −34.87 *** | −31.05 *** | −7.392 * | −10.08 ** | −1.726 | −3.555 |
| | (3.771) | (3.846) | (3.535) | (4.285) | (3.424) | (3.468) | (2.580) | (2.739) | (8.314) | (6.408) | (3.870) | (4.524) | (2.972) | (3.085) |
| Total Assets | | 0.222 | | 0.218 ** | | 0.400 * | | 0.226 ** | | −0.142 | | 0.254 | | 0.116 |
| | | (0.149) | | (0.103) | | (0.208) | | (0.108) | | (0.218) | | (0.235) | | (0.124) |
| ROA | | 1.882 * | | 2.124 | | 4.358 *** | | 0.268 | | −1.420 | | 1.782 ** | | 0.617 |
| | | (0.984) | | (1.298) | | (0.579) | | (1.877) | | (2.075) | | (0.857) | | (1.237) |
| NPM sector | | −0.0920 *** | | 0.0164 | | −0.143 *** | | −0.174 *** | | −0.149 *** | | −0.130 *** | | 0.0618 *** |
| | | (0.00667) | | (0.0174) | | (0.0186) | | (0.00982) | | (0.00900) | | (0.0260) | | (0.0158) |
| Inflation Rate | | −0.389 | | −0.871 | | −0.945 | | −0.441 | | 1.096 ** | | −0.966 | | −0.774 |
| | | (0.445) | | (0.828) | | (0.588) | | (0.473) | | (0.489) | | (0.734) | | (0.509) |
| GDP | | 0.143 | | 0.102 | | −0.196 * | | 0.0975 | | 0.452 | | 0.148 | | 0.180 |
| | | (0.129) | | (0.141) | | (0.105) | | (0.0983) | | (0.286) | | (0.140) | | (0.129) |
| G/GDP | | 0.343 ** | | 0.297 * | | 0.293 ** | | 0.231 * | | 0.679 *** | | 0.495 *** | | 0.382 *** |
| | | (0.134) | | (0.163) | | (0.127) | | (0.123) | | (0.245) | | (0.181) | | (0.121) |
| *Industry dummies (COMPUSTAT)* | YES | YES | YES | YES | YES | YES | YES | YES | YES | YES | YES | YES | YES | YES |
| Year dummies | YES | YES | YES | YES | YES | YES | YES | YES | YES | YES | YES | YES | YES | YES |
| Observations | 3801 | 3627 | 4194 | 3980 | 4194 | 3980 | 4194 | 3980 | 4194 | 3980 | 4194 | 3980 | 3801 | 3627 |
| Number of Countries | 26 | 26 | 26 | 26 | 26 | 26 | 26 | 26 | 26 | 26 | 26 | 26 | 26 | 26 |
| Number of Firms | 978 | 950 | 985 | 957 | 985 | 957 | 985 | 957 | 985 | 957 | 985 | 957 | 978 | 950 |

Robust standard errors clustered at country level are reported in parentheses. *** $p < 0.01$, ** $p < 0.05$, * $p < 0.1$. Omitted categories: Aerospace (Industry-COMPUSTAT); 2003 (Year); English Origins. NPM: Net Profit Margin at industry level; ROA: Return on assets.

## 6. Discussion and Conclusions

Corporate social responsibility is an emerging and growing phenomenon in contemporary globally integrated economies [49]. In spite of its increasing importance, there are fewer theoretical and empirical analyses on the impact that different legal origins may have on the implementation of CSR practices in the different CSR domains. Our paper aimed to bridge this gap by providing an original contribution to both the CSR and the legal origin literature.

We started by wondering whether the two different (civil and common) law traditions may have intrinsic characteristics that justify different patterns of the adoption of CSR practices. Based on the history of the two different cultures and on the established results in the literature, we formulated the hypothesis that the Anglo-Saxon tradition of corporate philanthropy could tilt the balance toward common law countries in the related CSR domain (community involvement). We argued that this may be the case since, as is well known, in the distribution of benefits from corporate action, common law is much more oriented toward shareholder protection, while civil law (especially in the French family) toward workers' rights.

Our descriptive and econometric findings document evidence (even not univocal) in the three indicated directions. More specifically, controlling for industry average characteristics and time fixed effects, the common law legal origin is positively and significantly correlated with the corporate governance and community involvement domains, while the French family in the civil law tradition is positively associated with the human resources domain (the CSR domain mostly concerning the workers' rights).

We finally observed no influence of legal origins on the environmental domain. We provide empirical evidence suggesting that this "non result" was the outcome of a remarkable process of convergence between the two legal origin groups. We further documented that convergence actually occurred in all domains but—consistently with Meyer's [42] theoretical predictions—it canceled out legal origin effects only in the environment and in the business behavior domains. We interpreted this last evidence in three ways. First, globalization reduces the influence of country-of-origin effects (producing convergence both within and between legal origin areas). Second, in some specific domains, such as that of environmental sustainability, the emergence of a global social norm (probably fostered by the creation and generalized voluntary adoption of some international standards) rapidly reduced differences among corporations coming from different legal cultures. Third, the increased use of benchmarking practices reinforces processes of creation of global social norms around commonly accepted environmental standards.

Overall, the policy implications of our results are that demand-driven pressure on social and environmental concerns, benchmarking practices, and global social norms are key factors that can increase corporate attention in contributing to the production of public goods and in addressing externalities. Hence, our findings suggest that the solution of global problems may not depend just on top-down institutional action but on the complex interplay of four forces, namely the market, institutions, active citizenship creating demand pressure on corporations, and corporate responsibility.

A limitation in our research is that we did not have instrumental variable estimates that could verify the causality nexus in the relationship between legal origins and corporate social responsibility. However, given that legal origins trace back in the past and can be hardly suspected of being determined by other drivers also affecting CSR, we are confident that our findings revealed a causality nexus. Nevertheless, further research in this direction is indeed welcome and would significantly contribute to enrich this field of the literature.

**Author Contributions:** Even though the work is the fruit of a common research we can divide the specific contributions as follows. Conceptualization, L.B., R.C. and P.C.; Methodology, P.C.; Formal Analysis, P.C.; Data Curation, R.C. and P.C..; Writing – Original Draft Preparation, L.B., R.C., P.C.; Writing – Review & Editing, L.B., R.C., P.C. All authors have read and agreed to the published version of the manuscript.

**Funding:** This research received no external funding.

**Conflicts of Interest:** The authors declare no conflict of interest.

## Appendix A

**Table A1.** Determinants of deviations from average industry CSR over time: random effects.

| Variables | (1) Overall Score | (2) Human Resources | (3) Environment | (4) Business Behavior | (5) Corporate Governance | (6) Community Involvement | (7) Human Rights |
|---|---|---|---|---|---|---|---|
| ln(Total Assets) | 0.0438 | 0.115 ** | 0.0646 | 0.119 ** | 0.104 | 0.130 | 0.0323 |
| | (0.0537) | (0.0581) | (0.0776) | (0.0565) | (0.0850) | (0.0983) | (0.0676) |
| Civil Law | −3.899 *** | −0.162 | −3.009 *** | −1.528 * | −14.05 *** | −4.490 *** | −0.853 |
| | (1.032) | (1.551) | (1.154) | (0.823) | (3.314) | (0.923) | (1.061) |
| GDP | −0.116 * | −0.00781 | −0.166 ** | −0.0457 | −0.354 | −0.227 *** | −0.00955 |
| | (0.0602) | (0.0772) | (0.0664) | (0.0549) | (0.218) | (0.0581) | (0.0586) |
| G/GDP | 0.0296 | 0.0938 | 0.113 | 0.0503 | −0.0725 | 0.00168 | 0.0960 |
| | (0.0649) | (0.108) | (0.0857) | (0.0522) | (0.122) | (0.0803) | (0.0749) |
| Industry dummies | NO | NO | NO | NO | NO | NO | NO |
| Year dummies | YES | YES | YES | YES | YES | YES | YES |
| Observations | 6757 | 7747 | 7747 | 7747 | 7747 | 7747 | 6757 |
| Number of Countries | 27 | 27 | 27 | 27 | 27 | 27 | 27 |
| Number of Firms | 1834 | 1,35 | 1835 | 1835 | 1835 | 1835 | 1834 |

Robust standard errors clustered at country level are reported in parentheses. *** $p < 0.01$, ** $p < 0.05$, * $p < 0.1$. Omitted categories: Aerospace (Industry); 2003 (Year).

**Table A2.** Determinants of deviations from average industry CSR over time: random effects.

| Variables | (1) Overall Score | (2) Human Resources | (3) Environment | (4) Business Behavior | (5) Corporate Governance | (6) Community Involvement | (7) Human Rights |
|---|---|---|---|---|---|---|---|
| ln(Total Assets) | 0.0511 | 0.128 ** | 0.0535 | 0.122 ** | 0.119 | 0.134 | 0.0356 |
| | (0.0547) | (0.0560) | (0.0805) | (0.0584) | (0.0823) | (0.103) | (0.0653) |
| French Origins | −2.668 | 2.386 | −4.210 ** | −0.763 | −12.50 *** | −2.823 * | −0.251 |
| | (1.742) | (2.358) | (1.852) | (1.181) | (4.061) | (1.620) | (1.551) |
| Scandinavian Origins | −4.230 ** | −1.907 | −4.929 ** | −2.379 | −10.06 *** | −8.776 *** | −0.959 |
| | (1.846) | (2.265) | (2.444) | (1.665) | (3.845) | (1.516) | (2.127) |
| German Origins | −4.391 *** | −0.918 | −1.897 | −1.668 * | −15.84 *** | −4.157 *** | −1.099 |
| | (1.071) | (1.567) | (1.362) | (0.957) | (3.238) | (0.969) | (1.344) |
| GDP | −0.115 * | 0.00734 | −0.141 ** | −0.0373 | −0.404 * | −0.178 *** | −0.00941 |
| | (0.0620) | (0.0751) | (0.0634) | (0.0560) | (0.218) | (0.0543) | (0.0605) |
| G/GDP | 0.000971 | 0.0475 | 0.173 ** | 0.0412 | −0.176 | 0.0190 | 0.0802 |
| | (0.0634) | (0.0957) | (0.0735) | (0.0546) | (0.136) | (0.0615) | (0.0657) |
| Industry dummies | NO | NO | NO | NO | NO | NO | NO |
| Year dummies | YES | YES | YES | YES | YES | YES | YES |
| Observations | 6757 | 7747 | 7747 | 7747 | 7747 | 7747 | 6757 |
| Number of Countries | 27 | 27 | 27 | 27 | 27 | 27 | 27 |
| Number of Firms | 1834 | 1835 | 1835 | 1835 | 1835 | 1835 | 1834 |

Robust standard errors clustered at country level are reported in parentheses. *** $p < 0.01$, ** $p < 0.05$, * $p < 0.1$. Omitted categories: Aerospace (Industry); 2003 (Year); English Origins.

**Table A3.** Determinants of deviations from average industry CSR: random effects (Mundlak correction).

| Variables | (1) Overall Score | (2) Human Resources | (3) Environment | (4) Business Behavior | (5) Corporate Governance | (6) Community Involvement | (7) Human Rights |
|---|---|---|---|---|---|---|---|
| ln(Total Assets) | −0.0258 | 0.859 ** | −0.0550 | 0.556 * | 1.086 ** | 0.562 | −0.641 |
| | (0.285) | (0.361) | (0.471) | (0.318) | (0.486) | (0.432) | (0.494) |
| Civil Law | −4.424 *** | −0.563 | −3.038 ** | −1.906 ** | −14.71 *** | −4.728 *** | −1.243 |
| | (0.912) | (1.590) | (1.245) | (0.868) | (2.861) | (0.916) | (1.161) |
| GDP | 0.217 | 0.138 | −0.0300 | 0.233 | 0.247 | 0.0788 | 0.331 |
| | (0.242) | (0.227) | (0.256) | (0.231) | (0.327) | (0.272) | (0.202) |
| G/GDP | 0.0208 | −0.0283 | 0.210 | 0.0505 | 0.0324 | 0.0915 | 0.101 |
| | (0.0748) | (0.152) | (0.151) | (0.0864) | (0.173) | (0.128) | (0.0817) |
| $\overline{\text{Total Assets}}$ | 0.0805 | −0.798 ** | 0.131 | −0.466 | −1.055 ** | −0.454 | 0.710 |
| | (0.307) | (0.406) | (0.505) | (0.343) | (0.484) | (0.482) | (0.535) |
| $\overline{\text{GDP}}$ | −0.360 | −0.156 | −0.146 | −0.293 | −0.632 * | −0.312 | −0.359 ** |
| | (0.222) | (0.205) | (0.238) | (0.198) | (0.331) | (0.244) | (0.174) |
| $\overline{\text{G/GDP}}$ | −0.00317 | 0.146 | −0.143 | −0.0220 | −0.188 | −0.137 | −0.0198 |
| | (0.0911) | (0.176) | (0.163) | (0.111) | (0.205) | (0.144) | (0.0856) |
| | −0.0258 | 0.859 ** | −0.0550 | 0.556 * | 1.086 ** | 0.562 | −0.641 |
| Industry dummies | NO | NO | NO | NO | NO | NO | NO |
| Year dummies | YES | YES | YES | YES | YES | YES | YES |
| Observations | 6757 | 6757 | 6757 | 6757 | 6757 | 6757 | 6757 |
| Number of Countries | 27 | 27 | 27 | 27 | 27 | 27 | 27 |
| Number of Firms | 1834 | 1834 | 1834 | 1834 | 1834 | 1834 | 1834 |

Robust standard errors clustered at country level are reported in parentheses. *** $p < 0.01$, ** $p < 0.05$, * $p < 0.1$. Omitted categories: Aerospace (Industry); 2003 (Year.

**Table A4.** Determinants of deviations from average industry CSR: random effects (Mundlak correction).

| Variables | (1) Overall Score | (2) Human Resources | (3) Environment | (4) Business Behavior | (5) Corporate Governance | (6) Community Involvement | (7) Human Rights |
|---|---|---|---|---|---|---|---|
| ln(Total Assets) | −0.0453 | 0.824 ** | −0.0241 | 0.553 * | 1.053 ** | 0.552 | −0.649 |
| | (0.296) | (0.377) | (0.479) | (0.319) | (0.487) | (0.437) | (0.510) |
| French Origins | −3.606 ** | 1.807 | −4.551 ** | −1.600 | −13.95 *** | −3.332 ** | −0.991 |
| | (1.504) | (2.112) | (1.854) | (1.125) | (3.709) | (1.566) | (1.400) |
| Scandinavian Origins | −4.542 *** | −2.355 | −4.794 ** | −2.564 | −9.844 *** | −8.695 *** | −1.177 |
| | (1.702) | (2.212) | (2.345) | (1.630) | (3.306) | (1.454) | (2.196) |
| German Origins | −4.685 *** | −0.974 | −2.056 | −1.852 * | −16.21 *** | −4.249 *** | −1.351 |
| | (1.084) | (1.627) | (1.422) | (1.034) | (2.807) | (1.040) | (1.455) |
| GDP | −0.0453 | 0.824 ** | −0.0241 | 0.553 * | 1.053 ** | 0.552 | −0.649 |
| | (0.296) | (0.377) | (0.479) | (0.319) | (0.487) | (0.437) | (0.510) |
| G/GDP | −3.606 ** | 1.807 | −4.551 ** | −1.600 | −13.95 *** | −3.332 ** | −0.991 |
| | (1.504) | (2.112) | (1.854) | (1.125) | (3.709) | (1.566) | (1.400) |
| $\overline{\text{Total Assets}}$ | −4.542 *** | −2.355 | −4.794 ** | −2.564 | −9.844 *** | −8.695 *** | −1.177 |
| | (1.702) | (2.212) | (2.345) | (1.630) | (3.306) | (1.454) | (2.196) |
| $\overline{\text{GDP}}$ | −4.685 *** | −0.974 | −2.056 | −1.852 * | −16.21 *** | −4.249 *** | −1.351 |
| | (1.084) | (1.627) | (1.422) | (1.034) | (2.807) | (1.040) | (1.455) |
| $\overline{\text{G/GDP}}$ | −0.0453 | 0.824 ** | −0.0241 | 0.553 * | 1.053 ** | 0.552 | −0.649 |
| | (0.296) | (0.377) | (0.479) | (0.319) | (0.487) | (0.437) | (0.510) |
| Industry dummies | NO | NO | NO | NO | NO | NO | NO |
| Year dummies | YES | YES | YES | YES | YES | YES | YES |
| Observations | 6757 | 6757 | 6757 | 6757 | 6757 | 6757 | 6757 |
| Number of Countries | 23 | 23 | 23 | 23 | 23 | 23 | 23 |
| Number of Firms | 1822 | 1822 | 1822 | 1822 | 1822 | 1822 | 1822 |

Robust standard errors clustered at country level are reported in parentheses. *** $p < 0.01$, ** $p < 0.05$, * $p < 0.1$. Omitted categories: Aerospace (Industry); 2003 (Year); English Origins.

**Table A5.** Legal Origins and CSR scores in all sustainability drivers.

| | Panel A | | | | | | | | | | |
|---|---|---|---|---|---|---|---|---|---|---|---|
| | HR1 | HR2 | HR3 | HR4 | HR5 | HR6 | HR7 | HR8 | | | |
| Civil law | 9.535 *** | −6.139 ** | 4.025 | 3.252 * | 2.194 | 2.705 * | −6.445 *** | 2.964 | | | |
| | ENV1 | ENV2 | ENV3 | ENV4 | ENV5 | ENV6 | ENV7 | ENV8 | ENV9 | ENV10 | ENV11 |
| Civil law | −3.616 | −0.0143 | 1.497 | −2.568 | −2.836 | 0.751 | −3.433 | 7.250 *** | −4.082 | 0.537 | −4.009 * |
| | CS1 | CS2 | CS3 | CS4 | CS5 | CS6 | CS7 | CS8 | CS9 | CS10 | |
| Civil law | 5.054 *** | −4.077 *** | −7.143 *** | −1.013 | −2.081 | −0.967 | −2.601 | −3.368 * | 0.305 | −0.422 | |

**Table A5.** *Cont.*

### Panel A

|  | CG1 | CG2 | CG3 | CG4 |
|---|---|---|---|---|
| Civil law | −26.60 *** | −19.34 *** | −19.66 *** | −35.41 *** |

|  | CIN1 | CIN2 | CIN3 |
|---|---|---|---|
| Civil law | −4.468 *** | −6.011 *** | −12.30 *** |

|  | HRT1 | HRT2 | HRT3 |
|---|---|---|---|
| Civil law | 1.068 | 2.960 *** | −3.920 ** |

### Panel B

|  | HR1 | HR2 | HR3 | HR4 | HR5 | HR6 | HR7 | HR8 |
|---|---|---|---|---|---|---|---|---|
| French Origins | 15.18 *** | −6.555 | 6.843 ** | 8.163 *** | 8.483 ** | 3.588 | −1.682 | 1.986 |
| Scandinavian Origins | 6.793 * | −0.634 | −7.930 * | −0.372 | −1.028 | 3.291 | −6.527 ** | −2.996 |
| German Origins | 6.569 * | −5.593 | 0.0958 | 1.595 | −0.742 | 1.739 | −9.213 *** | 4.269 ** |

|  | ENV1 | ENV2 | ENV3 | ENV4 | ENV5 | ENV6 | ENV7 | ENV8 | ENV9 | ENV10 | ENV11 |
|---|---|---|---|---|---|---|---|---|---|---|---|
| French Origins | −4.681 | 5.501 | 2.282 | −0.711 | 6.391 | −3.542 | 10.66 *** | −4.032 | 7.125 ** | −6.705 * | −3.746 |
| Scandinavian Origins | −4.162 | 0.574 | −3.848 | −3.878 | 1.994 | −5.774 * | 7.010 ** | −3.807 | −3.782 | −6.872 * | −5.549 * |
| German Origins | −2.822 | −2.918 * | 2.192 | −3.537 * | −2.288 | −2.885 | 5.669 *** | −4.179 | −1.338 | −1.898 | −1.092 |

|  | CS1 | CS2 | CS3 | CS4 | CS5 | CS6 | CS7 | CS8 | CS9 | CS10 |
|---|---|---|---|---|---|---|---|---|---|---|
| French Origins | 6.316 ** | −2.964 | −2.324 | 0.950 | −1.885 | 1.901 | −1.153 | −1.238 | −0.244 | −0.185 |
| Scandinavian Origins | 3.951 | −3.376 | −13.71 *** | −3.298 | −1.301 | 0.645 | −4.510 * | −4.666 | −1.005 | −3.113 ** |
| German Origins | 4.608 *** | −4.810 *** | −8.361 *** | −1.922 | −2.405 | −2.889 ** | −3.071 | −4.393 ** | 0.802 | −0.181 |

|  | CG1 | CG2 | CG3 | CG4 |
|---|---|---|---|---|
| French Origins | −25.68 *** | −10.39 *** | −22.41 *** | −35.19 *** |
| Scandinavian Origins | −11.33 ** | −21.15 *** | −30.48 *** | −36.69 *** |
| German Origins | −30.26 *** | −24.07 *** | −16.07 *** | −35.31 *** |

|  | CIN1 | CIN2 | CIN3 |
|---|---|---|---|
| French Origins | −0.483 | −2.545 | −11.67 *** |
| Scandinavian Origins | −9.666 *** | −15.85 *** | −21.77 *** |
| German Origins | −5.349 *** | −6.356 *** | −11.50 *** |

|  | HRT1 | HRT2 | HRT3 |
|---|---|---|---|
| French Origins | −0.503 | 6.063 *** | −2.061 |
| Scandinavian Origins | 2.461 | 3.806 ** | −5.582 |
| German Origins | 1.946 | 1.097 | −4.720 *** |

Legend: HR1: Promotion of labor relations; HR2: Encouraging employee participation; HR3: Training and development; HR4: Responsible management and restructurings; HR5: Career management and promotion of employability; HR6: Quality of remuneration systems; HR7: Improvement of health and safety conditions; HR8: Respect and management of working hours; ENV1: Environmental strategy and eco-design; ENV2: Pollution prevention and control; ENV3: Development of green products and services; ENV4: Protection of biodiversity; ENV5: Protection of water resources; ENV6: Minimizing environmental impacts from energy use; ENV7: Management of atmospheric emissions; ENV8: Waste management; ENV9: Management of environmental nuisances: dust, odor, noise; ENV10: Management of environmental impact from transportation; ENV11: Management of environmental impact from the use and disposal of products/services; CS1: Product safety; CS2: Information customers; CS4: Responsible contractual agreement; CS3: Sustainable relationship with supplies; CS4: Integration of Environmental factors in the supply chain; CS5: Integration of social factors in the supply chain; CS6: Prevention of corruption; CS7: Prevention of anti-competitive practices; CS8: Transparency and integrity of influence strategies and practices; CG1: Board of Director; CG2: Audit and Internal Control; CG3: Shareholders Rights; CG4: Executive Remuneration; CIN1: Promotion of social and economic development; CIN2: Social impacts of company products and services; CIN3: Contribution to general interest causes; HRT1: Respect for human rights standards and prevention of violations; HRT2: Respect for freedom of association and their right to collective bargaining; HRT3: Non-discrimination. *** $p < 0.01$, ** $p < 0.05$, * $p < 0.1$.

**Appendix B  VIGEO Rating Domains and Sustainability Drivers.**

<u>**Human Resources.**</u> **Promotion of labor relations**: company's commitment to ensuring the respect of independent workers' representatives through information, consultation, and notably collective bargaining, at the workplace. **Encouraging employee participation**: company's commitment to defend and promote employees' individual information and expression, and employees' participation in decision-making on matters not related to collective bargaining. **Responsible management of restructurings**: capability to inform and consult employee representatives before/during restructuring process, to put in place practical measures, to prevent and limit redundancies (notably budgets, processes, and reporting), and to take measures to mitigate the negative effects of redundancies on employees, notably reemployment measures. **Career management and promotion of employability**: company efforts to anticipate short- and long-term employment needs and skill requirements, adapt employees' skill sets to their career paths, enable the progressive improvement in employees' qualification levels, and put in place a concerted career management framework, which is transparent and individualized. **Quality of remuneration systems**: company's commitment to ensure the decency, transparency, and objectivity of employees' remuneration systems. **Improvement of health and safety conditions**: company's commitment regarding the protection of employees' health and safety. **Respect and management of working hours**: initiatives taken by the company to promote the voluntary flexibility of working hours.

<u>**Environment.**</u> **Environmental strategy and eco-design**: company's commitment to define clear objectives and appropriate measures to ensure management of the environmental impacts of products and services. **Pollution retention and control**: extent to which the company is preventing and managing risks of accidental pollution or soil pollution. **Development of green products and services**: company's efforts to develop: i) products and services with significantly decreased environmental impact; and ii) that may be considered as a fundamental diversification for the enterprise, either at the level of the production process (wind turbine for electricity producers), or at the product (hydrogen for oil producers or fuel cells for car makers) or at service level (green investment funds in banking sector). **Protection of biodiversity**: company's commitment to prevent risks of endangering biodiversity. Company's commitment to manage animal testing (when relevant for the sector). **Protection of water resources**: measures taken to reduce water consumption and to improve, reduce, or treat wastewater emissions/water discharges. **Minimizing environmental impacts from energy use**: company's efforts to address and minimize energy-related issues (energy consumption and emissions related to energy consumption). **Management of atmospheric emissions**: steps taken by the company to control atmospheric emissions related to the production of products/projects/services. Atmospheric emissions resulting from the company's energy consumption are out of the scope of this criterion, see: 2.2—Minimizing environmental impacts from energy use and related atmospheric emissions. **Waste management:** Steps taken by companies to manage waste: i) identification of the different sources of waste; ii) reduction of waste production at source; iii) management of industrial and commercial packaging and packaging waste; iv) waste recycling, energy recovery from waste (waste to energy); v) reduce the toxicity of hazardous waste. **Management of environmental nuisances: dust, odor, noise (Management of local pollution)**: company management and reduction of local pollution (noise, dust, and odors) resulting from the production processes and maintenance of installations, as well as local degradation of the environmental aesthetics. **Management of environmental impact from transportation**: company effort and results when taking into account environmental impact of its products' transportation and actions that are implemented to reduce these impacts.

<u>**Business Behavior.**</u> **Product safety**: corporate attention to product safety issues into account, and the related steps taken to prevent and repair emergency/crisis situations affecting product safety. **Information customers**: definition and implementation of principles of conduct and measures to prevent negative impact of marketing practices on financial, moral, and ethical issues as well as on the health and safety of users and/or customers. **Responsible contractual agreement**: corporate commitment to include guarantees in its contractual relationship, which promote customers' freedom

of decision, satisfaction, and right to recourse. **Sustainable relationship with suppliers**: corporate commitment to ensure balanced and sustainable relationships with suppliers, focusing on: i) promoting mutually beneficial business relations; ii) optimizing mutual profits gained through contract in terms of quality, costs, and technical/technological control. **Integration of environmental factors in the supply chain**: Evaluation of the extent to which the company integrates environmental factors in the supply chain. **Integration of social factors in the supply chain**: Evaluation of the extent to which the company is integrating social standards into supply chain. **Prevention of corruption**: effectiveness of the company's anti-corruption management system. Corruption is studied in its broadest sense. Conflicts of interest are also taken into account, as they can cast doubt on the quality of the company decision-making process and on the integrity of people involved. **Prevention of anti-competitive practices**: corporate consideration for competition laws and the prevention of market distortion rules in its relationships with customers, suppliers, and competitors. **Transparency and integrity of influence strategies and practices**: corporate disclosure of the objectives of its lobbying practices and the resources dedicated to achieving them. Appointment of clear responsibilities and designation of specific procedures to monitor the correct implementation of the company's lobbying strategy.

**Corporate Governance. Board of Directors**: corporate commitment to set up a board of directors that is capable of controlling and advising executives and that is held accountable to shareholders. **Audit and Internal Control**: corporate commitment to establish effective risk management systems, ensuring the quality of internal reporting, and the extent to which this commitment is reflected in financial information provided to the public. The board of directors is responsible for the objectivity and relevance of the system. **Shareholders Rights**: corporate commitment to ensure the fair treatment of shareholders, allowing them to actively participate in strategic decision-making. Voting rights attached to shares and the right to participate in general meetings are of fundamental importance in this regard. **Executive Remuneration**: corporate commitment to use executive remuneration as a tool to align the interests of executives and shareholders.

**Community Involvement. Promotion of social and economic development**: corporate commitment to provide sustainable contributions to the economic and social development of local areas and to optimize the economic and social impact of activities: local investment, promotion of local employment, transfer of technologies and skills. **Social impacts of company products and services**: development of voluntary initiatives taking into account their product or services' impact on the community. **Contribution to general interest causes**: corporate commitments to promote voluntary community initiatives not directly related to the company's products or services: patronage, involvement in various causes of general interest, other forms of sponsorship, as well as contributions to studies or academic research on community interest issues.

**Human Rights. Respect for human rights standards and prevention of violations**: extent to which the company is complying with obligation to respect human rights in the community (community taken as a whole, i.e., within and outside of the workplace). This obligation includes: respect of effective exercise of fundamental human rights and personal rights; prevention of human rights violations or complicity of violations. **Respect for freedom of association and their right to collective bargaining**: respect of trade union freedom, collective bargaining rights, and promotion of collective bargaining rights. **Elimination of child and forced labor**: corporate contribution to the elimination of child labor and/or forced labor. **Non-discrimination**: corporate prevention of gender discrimination on workplace and other discrimination regarding work conditions, vocational training, promotion, fees, and other benefits. Positive measures and specific measures intended to protect and support women (pregnancy, maternity) or vulnerable people, constitute measures to promote equal opportunity and treatment.

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
