# Peer review of "Legal Origins and Corporate Social Responsibility"

_sustainability, doi:10.3390/su12072717_

Round 1

Reviewer 1 Report

The paper titled “The Legal Origins of Corporate Social Responsibility” presents an interesting paper, with an interesting approach and subject of the research. Although the observed period of research is from 2007 to 2013 the research results are relevant and do not, in my opinion, greatly affect the research results. The title of the paper is however somehow misleading and I advise to adjust it in order to reflect the effects of prevalent legal framework on CSR domains, which is actually the subject of the paper. In addition, several suggestion are the following ones:

  • In their introduction, authors provide data referring to and increasing corporate practice in the most recent years. However, data refer to 2011, and as such are a bit outdated. So please, incorporate some newer data.
  • Reconsider if providing reasoning for each hypothesis would be better before you state your hypotheses. Because, as I was reading your paper, I personally lacked more clear reasoning behind your hypothesis. Namely, why did you presume that common law countries will have higher CSR scores in the Community Involvement domain; and Civil law countries (and, more specifically, the French tradition) will have higher scores in the CSR labor domain (Human Resources)? From your paper, it was hard to detect the clear reasoning for these two assumptions based on literature surveyed and presented in the section Legal Origin Culture and Stakeholder Rights. Only after, as I read your paper I understood the assumptions and grounds of your hypotheses. So just, reconsider if changing the order of the text would be better to ensure clear flow of the text for the readers.
  • I do not feel qualified to assess the econometric model, but the presentation of the results is good and methodological aspects of the paper seem to be appropriate.
  • Conclusion is lacking practical and theoretical implications, as well as research limitations.

Author Response

Dear reviewer 

please find enclosed the file with our replies to all the comments and suggestions received 

Reviewer 2 Report

The abstract should be rewritten, it is unclear and vague. What is the novelty of this paper? The aim and policy implication of this paper is not clear. The cited references are very old (only one paper published in 2016), which shows that the topic is not interesting. The used data is very old (2003-2013), which shows the lack of novelty. I do not agree that Russia could belong to German civil law. How were measured these aspects as: community involvement, ENvironment and Bussiness behavior? Econometric methods should be presented. The discussion section was not presented. What is the policy implication of this paper? 

Author Response

Dear reviewer

please find enclosed the file with our response and changes related to all the points raised

Regards

Round 2

Reviewer 2 Report

I would like to ask to revise and renew references, because they are very old.

Author Response

REVIEWER'S POINT

I would like to ask to revise and renew references, because they are very old.

Dear Reviewer

We follow the advice and add four contributions belonging to the subsection of papers on our topic (legal origins and CSR) with more recent date than 2016.

References

Kim, Hakkon, Kwangwoo Park, and Doojin Ryu. "Corporate environmental responsibility: A legal origins perspective." Journal of Business Ethics 140.3 (2017): 381-402.

Liang, Hao, and Luc Renneboog. "On the foundations of corporate social responsibility." The Journal of Finance 72.2 (2017): 853-910.

Jansen, Joëla. "The effect of corporate social responsibility on the cost of equity from a legal origin and cultural perspective." (2017).

Ang, James B. "Culture, legal origins, and financial development." Economic Inquiry 57.2 (2019): 1016-1037.